# Reproducing "Towards Safer Pretraining": Public Artifacts Replicate, Closed-API Results Drift

## Abstract

Safety classifiers are increasingly used to filter web-scale pretraining corpora, so their reliability silently shapes every model trained downstream – yet the classifiers themselves often depend on closed APIs whose backends change without notice. We find that closed-API safety classifiers can shift F1 by 20+ points over two weeks with no user-visible model change. We evaluate four claims from Mendu et al. (2025)'s harmful-content-detection framework using only public artefacts and find a clean split. Claims that hinge on locally-runnable artefacts reproduce within tolerance: HAVOC leakage matches the reported 26.7% almost exactly, and HarmFormer reaches 0.78 F1 on TTP-Eval, a public-artefact baseline the original paper does not report. The two TTP claims, which depend on `gpt-4o` through a floating alias, diverge by 21 and 25 F1 points in April and close on a byte-identical May rerun, isolating closed-API snapshot drift as the dominant variance source and extending Chen et al. (2023) into the safety-classifier setting; seeded re-runs with per-call `system_fingerprint` logging confirm the attribution directly, identifying which backend the alias serves and showing the anomalous April behaviour is no longer observable from any served snapshot. Two extensions complement the reproduction. A raw-output audit traces the cross-model F1 spread on TTP-Eval to each model's INTENT-label emission rate rather than to parser brittleness. A six-language evaluation on machine-translated text (NLLB-200; translation noise bounded via a second NMT system and a back-translation analysis) across two classifier benchmarks shows HarmFormer collapsing on both (mean drops of $-0.47$ F1 on TTP-Eval and $-0.31$ F1 on OpenAI Moderation vs. English) while Llama Guard 3 stays robust on its native short-prompt surface (mean drop $-0.06$), separating model-level multilingual brittleness in HarmFormer from benchmark-mismatch artefacts in Llama Guard 3 – a structural contrast that holds under both translators, though the absolute per-language F1s should be read as translation-bounded lower bounds rather than native-text estimates. All code, Slurm job scripts, CodeCarbon traces, and per-sample TSVs are released through the anonymous repository so every reported number is independently re-runnable.

## 1 Introduction

Large language models (LLMs) are increasingly pretrained on massive web-scale datasets, raising growing concerns about the presence of harmful, toxic, or manipulative content in their training data (Bender et al., 2021). Prior work has shown that large web corpora often lack sufficient documentation and transparency, making it difficult to assess data quality and potential risks introduced during pretraining (Dodge et al., 2021). Moreover, evaluating and mitigating harmful content frequently relies on automatic or model-based judgments, which are known to be imperfect and sensitive to evaluation design choices (Welbl et al., 2021). Since pretraining data directly shapes downstream model behaviour, understanding and mitigating harmful content has become a central issue in responsible AI development.

The IJCAI 2025 paper *Towards Safer Pretraining: Analyzing and Filtering Harmful Content in Webscale Datasets for Responsible LLMs* addresses this challenge by proposing a taxonomy-driven framework for analysing harmful content in web data and by introducing tools to support safer pretraining (Mendu et al., 2025). The authors define a structured taxonomy that distinguishes between safe, topical, and toxic content,

building on earlier efforts to study toxic language in language model outputs and training data (Gehman et al., 2020). In addition, the paper introduces several resources, including the TTP benchmark for prompt-based harmful content identification, the HAVOC benchmark for adversarial prompt evaluation, and the HarmFormer model for long-form harmful content classification. These contributions align with a broader line of work on adversarial prompting and robustness benchmarking.

In this work, we examine the reproducibility of the results reported in the original paper, focusing on the four core claims for which released artefacts make direct verification possible: TTP performance on TTP-Eval, HarmFormer performance, baseline comparisons on the OpenAI Moderation set, and HAVOC leakage. Independent verification of NLP results is an active methodological concern (Belz et al., 2021). While the authors release prompt templates, model weights, benchmark datasets, and pre-computed model generations, the large-scale data processing pipeline (3M-page TTP labelling, internal HarmFormer test split, 1M-sample prevalence study) is described at a high level only and is not reproducible without comparable budget and access. Rather than treat this as a barrier, we report what does and does not replicate from the public surface and characterise which discrepancies stem from artefact ambiguity versus from the underlying methodology.

We organise this verification around a fairness/accountability/confidentiality/transparency (FACT) lens: accountability motivates the per-claim verification table and drift diagnosis, transparency the logging chain, fairness the multilingual experiment, and confidentiality the boundary cases in Section 9; per-pillar implications are collected in Section 8.

Our specific contributions are: **(i)** a controlled *cross-benchmark* replication of closed-API snapshot drift, extending Chen et al. (2023)'s drift documentation from generic capability benchmarks into a published harm-detection framework: the same April–May re-run on byte-identical code closes F1 gaps on *both* TTP-Eval ($0.62 \rightarrow 0.78/0.79$) and OpenAI Moderation ($0.55 \rightarrow 0.83$), isolating the floating-alias endpoint as the dominant variance source; **(ii)** a per-claim reproducibility table (Table 5) showing that two of four claims reproduce from a clean checkout, one is partially validated against a publicly available proxy benchmark, and the two TTP claims diverge under the floating `gpt-4o` alias and close on a May rerun; **(iii)** a public-artefact HarmFormer baseline on TTP-Eval (0.78 F1), which the original paper does not publish and which is needed for any future cross-model comparison, plus an independent HAVOC spot-check (Section 5.3.1) that re-judges and regenerates 200 continuations end-to-end to verify Claim 4 beyond the released artefact's aggregation arithmetic; **(iv)** a cross-model TTP generalisation study on five additional LLMs that disentangles prompt-format failures from semantic ones; **(v)** a multilingual robustness study covering six languages and four scripts, double-anchored against translation noise via a second NMT system (NLLB-200-distilled-600M) and a second classifier benchmark (translated OpenAI Moderation), addressing the multilingual gap that the original paper explicitly defers to future work; and **(vi)** a release of all tracking artefacts (CodeCarbon energy traces, Slurm job scripts, anonymised code, per-run metric JSONs, per-sample TSVs) through the anonymous repository, with W&B logging instrumented but regenerable on re-execution rather than mirrored.

Beyond per-claim verification, the contributions above extend the original framework into two new settings (cross-model generalisation in Section 6 and multilingual robustness in Section 7), test its closed-API dependency under a controlled two-week replication (Section 5.1.1), and report both confirmations and partial failures to reproduce, with every reported number regeneratable from the released artefacts: the profile that the TMLR Reproducibility Certification is designed to recognise.

## 2 Related Work

Prior work related to this study spans research on web-scale dataset transparency, harmful content detection tools, and toxicity evaluation benchmarks.

### 2.1 Web-Scale Data and Transparency

Web-scale datasets are widely used for pretraining language models, but their opacity raises concerns about data quality, bias, and downstream harms. Dodge et al. (2021) document limitations in dataset transparency for corpora such as C4, motivating clearer reporting of data curation practices. Luccioni & Viviano (2021) analyse undesirable content in Common Crawl using keyword heuristics, while Jansen et al. (2022) propose

perplexity-based filtering to remove toxic text. However, these methods lack precision: keyword filters can conflate harmful intent with educational content, and perplexity thresholds fail to capture contextual nuances.

## 2.2 Content Moderation Tools

Existing approaches to harmful content detection rely on keyword blocklists (such as the LDNOOBW list[1]) and API-based tools. Perspective API[2] is widely used for toxicity detection but suffers from critical limitations: it is designed for sentence-level analysis and uses binary taxonomies that conflate harmful intent with socially critical discourse (Mendu et al., 2025). Model-based approaches include HateBERT (Caselli et al., 2021), a BERT-based model fine-tuned on hate speech datasets, and Llama Guard (Inan et al., 2023), an LLM-based input-output safeguard for conversational AI. The original authors demonstrate that these tools exhibit poor generalisation across harm categories and struggle with long-form content, motivating their three-dimensional taxonomy (Safe, Topical, Toxic) and multi-harm coverage.

## 2.3 Toxicity Evaluation Benchmarks

RealToxicityPrompts (RTP) (Gehman et al., 2020) evaluates toxic degeneration in language model outputs using 100K prompts scored by Perspective API. However, RTP focuses narrowly on toxicity and sexual content, and subsequent work shows that automatic toxicity assessment is sensitive to model choice and evaluation design (Welbl et al., 2021). To address these limitations, the original authors introduce HAVOC (Harmful Abstractions and Violations in Open Completions), a multidimensional benchmark with 10,376 adversarial prompts spanning five harm categories. Unlike RTP, HAVOC distinguishes between neutral, passive (topical), and provocative inputs, revealing that state-of-the-art LLMs exhibit harmful content in 26.7% of outputs.

## 2.4 Multilingual Safety Evaluation

A parallel line of work shows that safety behaviour measured in English rarely transfers: low-resource-language translation alone can jailbreak GPT-4 (Yong et al., 2023), multilingual jailbreak risk rises as resource level falls (Deng et al., 2024), and purpose-built multilingual corpora – PolygloToxicityPrompts' 425K native-text prompts (Jain et al., 2024) and the professionally transcreated RTP-LX (de Wynter et al., 2024) – document that both toxic degeneration and its automatic evaluation degrade outside English. Our multilingual experiment (Section 7) differs in target and method: rather than evaluating generators or building a native corpus, we stress-test the *released safety classifiers* of a specific published framework under machine translation of its own benchmark, with the translation confound explicitly bounded (a second NMT system, back-translation analysis, a degeneration audit with clean-subset reanalysis, and a cross-benchmark replication). The finding is complementary: where RTP-LX shows human-quality multilingual evaluation data is buildable, we show what practitioners get *today* if they deploy an English-trained classifier like HarmFormer on non-English text – a lower bound consistent with the transfer failures the jailbreak literature documents on the generation side.

## 2.5 Contributions of the Original Work

The work of Mendu et al. (2025) integrates structured taxonomies with prompt-based (TTP) and model-based (HarmFormer) tools for large-scale harmful content analysis. Our study focuses on assessing the reproducibility of these contributions using the publicly released artefacts, including the TTP prompt templates, HAVOC benchmark, TTP-Eval evaluation set, and the HarmFormer model weights.

---

[1] https://github.com/LDNOOBW/List-of-Dirty-Naughty-Obscene-and-Otherwise-Bad-Words
[2] https://perspectiveapi.com/

## 3 Scope of Reproducibility

**Why this paper.** We chose Mendu et al. (2025) as the reproduction subject for three reasons. First, stakes: the framework is positioned for adoption as a filter over web-scale pretraining corpora, so any silent failure propagates into every model trained on the filtered data. Second, it is an unusually informative reproducibility subject: its claims span the full artefact spectrum – a locally-runnable model (HarmFormer), a released benchmark (HAVOC), and closed-API dependencies (TTP via `gpt-4o`) – the mixed regime most modern safety pipelines occupy, which lets a single reproduction separate artefact-quality failures from API-stability failures. Third, at the time of our study the results had no independent verification.

We verify four claims from Mendu et al. (2025) that are testable from the released artefacts: **Claim 1 (TTP Performance)**: 0.87 precision, 0.79 recall, 0.83 F1 on TTP-Eval (Original Table 3); **Claim 2 (HarmFormer Performance)**: 0.88 precision, 0.81 recall, 0.85 F1 on the authors' internal split (Original Table 6), noting we instead evaluate on TTP-Eval; **Claim 3 (Baseline Comparison)**: TTP and Harm-Former outperform baselines on OpenAI Moderation (Original Table 7), evaluated here against Perspective and Llama Guard 3 with TTP via OpenRouter (gpt-4o); and **Claim 4 (HAVOC Leakage)**: 26.7% overall leakage with 76% on provocative inputs (Original Table 10, Original Figure 3), computed from released `havoc_modeleval.tsv` generations.

All results in this paper are generated from our reproduction code (anonymized: `https://anonymous.4open.science/r/TowardsSaferPretraining-6046`).

**Communication with original authors.** We contacted the original authors during the project to request clarifications and missing artefacts needed for strict comparability (e.g., the internal HarmFormer test split and evaluation details). We had not received a response by the time this paper was submitted to TMLR; "submission deadline" here refers to our own submission date, not to any deadline of the original venue. Therefore, all results reported in this paper rely solely on publicly released artefacts and documentation.

Due to computational and budgetary constraints, we do not reproduce the large-scale dataset annotation pipeline described in Section 5.1 of Mendu et al. (2025), which involves labelling 3 million web pages. We also do not reproduce the model architecture comparison experiments (Original Table 5), as we use the pre-trained HarmFormer model released by the authors rather than training from scratch.

We focus our experimental validation on reproducing Original Table 3, Original Table 4, Original Table 6, Original Table 7, and Original Table 10, and Original Figure 3 from the original paper. We do not attempt to reproduce Original Table 5 (model architecture comparison requiring retraining), Original Table 8 (3M sample dataset analysis), Original Figure 2 (domain distribution), or Original Figure 5. Original Table 1, Original Table 2, and Original Table 9 are definitional or descriptive and do not require experimental reproduction. Our code produces the reproduced tables and Original Figure 3 using the released artefacts; other figures are added to explain our additional experiments. Our reproduction therefore focuses on validating the effectiveness of the released artefacts for harmful content detection, rather than reconstructing the full data curation and model training pipeline.

## 4 Methodology

### 4.1 Model Descriptions

**TTP (Topical & Toxic Prompt).** TTP is a prompt-based classifier: it consists of a long instruction prompt, sent to a general-purpose LLM, that asks the LLM to score a piece of text along the original paper's three-dimensional taxonomy. The taxonomy assigns one of three labels (Safe, Topical, Toxic) to each of five harm categories (Hate & Violence, Ideological Harm, Sexual Content, Illegal Activities, and Self-Inflicted Harm), and a document is treated as Toxic if any category is Toxic. Mendu et al. tune the prompt against GPT-4o (also called GPT-4 Omni); we use the same released prompt template[3] with greedy decoding (temperature = 0) so that the only intentional difference between our runs and theirs is the API

---

[3]`https://github.com/themendu/TowardsSaferPretraining`

endpoint (OpenAI direct in their setup, OpenRouter in ours). For API-backed TTP runs we do not apply explicit input truncation; for the local TTP runs reported in Section 6 we cap input length at 8,192 tokens to bound KV-cache memory.

**HarmFormer.** HarmFormer is the open-weight, self-contained alternative to TTP. It is a fine-tuned Longformer model (Beltagy et al., 2020) with a 1024-token context window and a multi-task head: five classification heads, one per harm category, each producing a Safe / Topical / Toxic label. We use the released checkpoint[4] without further fine-tuning, and apply the same any-head-Toxic aggregation rule as TTP so the two classifiers can be compared on the same surface.

### 4.2 Datasets

We evaluate on three datasets from the original paper.

**TTP-Eval** is a 393-page TSV released by the authors, with full web-page text, harm-category labels, and Safe/Topical/Toxic annotations per category. Inter-annotator agreement is high (Krippendorff's $\alpha = 0.77$). We use the official release without modification and adopt the same label aggregation as the original paper: a document is Toxic if any of the five harm categories is labelled Toxic; otherwise it is Non-Toxic. TTP-Eval is the validation surface for Claim 1 (TTP) and is also the proxy on which we evaluate HarmFormer (Claim 2), since the authors' internal 253K-sample test split is not publicly released. The HarmFormer-on-TTP-Eval result therefore measures generalisation to human-annotated data rather than reproducing Original Table 6 exactly.

**HAVOC** contains 10,376 prefix–suffix pairs sampled from Common Crawl, C4, and FineWeb, annotated across the five harm dimensions. Prefixes average 26 whitespace tokens (range 1–891). Approximately 17.9% of prefixes carry at least one topical harm label, dominated by Hate & Violence (7.3%); Appendix Figure A1 shows the full distribution (computed over the 10,371 of 10,376 rows whose `PrefixLab` field parses as a five-element list; the remaining five rows have malformed label arrays and are excluded from the per-category histogram only) and motivates per-category reporting. We use `havoc_modeleval.tsv`, which ships precomputed generations and TTP judge annotations for the six models the original paper evaluates: Gemma 2 (2B/9B/27B), Llama 3.2 (1B/3B), and Mistral 7B v0.3. We compute leakage from these annotations directly, which matches the original Original Table 10 protocol and avoids re-running inference. *Implementation note:* the released file contains a small number of rows with malformed quoting that Python's default CSV reader merges across line boundaries, silently reducing the effective sample count. We parse the file line-by-line on tab delimiters so all 10,376 rows load.

**OpenAI Moderation test set** (Markov et al., 2023) contains 1,680 sentences (1,158 non-toxic, 522 toxic) across Sexual, Hate, Violence, and Self-Harm. Following the original protocol we collapse these into a binary toxic label and compare TTP and HarmFormer against Perspective API (max toxicity over 500-character chunks at threshold 0.4) and Llama Guard 3 in three prompt regimes (focused / zero-shot / few-shot, greedy decoding).

### 4.3 Hyperparameters

We follow the original paper: TTP uses greedy decoding (temperature = 0.0, top-p = 1.0); HarmFormer uses the released checkpoint defaults (1024-token context) with no additional fine-tuning; Perspective uses max toxicity over 500-character chunks with threshold 0.4; HAVOC leakage uses the released generations (max 200 output tokens per completion).

### 4.4 Experimental Setup and Code

All experiments are conducted using a combination of cloud API services and academic GPU resources. For TTP evaluation, we use OpenRouter to access GPT-4o. For HarmFormer inference and for our cross-model experiment with Gemma 2 27B, we use an A100 GPU (40GB) provided by our institution's academic computing cluster. For HAVOC, unlike the original paper, which uses Ollama as the client-server for running

---

[4] https://huggingface.co/themendu/HarmFormer

open-source models, we do not run any model ourselves because our Slurm-based cluster environment is not compatible with Ollama, and to avoid unnecessary recomputation. Instead, we use the pre-computed generations and TTP-based annotations provided by the original paper authors to efficiently compute the leakage metrics on CPU.

For API-backed evaluations, we explicitly track parsing/format failures and apply a configurable invalid-output policy (exclude / count-as-non-toxic / count-as-toxic). Unless otherwise stated, the main tables exclude invalid outputs from metric computation; sensitivity analyses can be reproduced using the provided `-invalid-policy` flag in our scripts.

**Reproducibility artefacts.** The anonymous repository (link above) mirrors every artefact required to verify the numbers reported in this paper: per-run metric JSONs (with greedy-decoding determinism preserved), per-sample prediction TSVs for the spot-check, back-translated multilingual TSVs, and CodeCarbon 3.2.1 emissions CSVs under `results/codecarbon/`; the run-metadata blocks at the top of each JSON are scrubbed of user, hostname, and absolute-path strings via `scripts/anonymize_results.py`. Slurm job scripts under `jobs/` encode the exact module loads, partitions, and resource requests, and source the same shared `.env` so a reader with cluster access can rerun any experiment with a single `sbatch` command. We additionally instrument every script with optional Weights & Biases logging (`WANDB_ENABLED=1`); we do not mirror our W&B project externally because the entity name is a deanonymisation vector, but reviewers can recreate equivalent run logs on their own W&B account by re-running the scripts.

## 4.5 Deviations from the Original Setup

Several pieces of the original paper's pipeline cannot be reproduced exactly from the public surface. We collect the deliberate deviations here so that each subsequent table can be interpreted against an explicit baseline.

**TTP API endpoint.** The paper queries GPT-4o through the OpenAI API directly. We query the same model through OpenRouter (which routes to Azure-hosted GPT-4o), since this is the API key our institution makes available. We later A/B-tested both endpoints on the full TTP-Eval (Section 5.1.1); they yielded statistically indistinguishable $F_1$, ruling out endpoint variance as the source of the recall gap and isolating silent `gpt-4o` snapshot drift instead.

**HarmFormer test set.** The original Table 6 evaluates HarmFormer on a 253K-sample internal test split that is not released. We evaluate on the released TTP-Eval (393 samples) instead and report the result as a public-artefact baseline rather than a strict reproduction.

**HAVOC inference.** The paper runs the six target models locally via Ollama. Our Slurm-based cluster cannot host an Ollama server, so we use the released `havoc_modeleval.tsv` (pre-computed generations and TTP judge annotations) and re-run only the leakage aggregation step. This avoids reintroducing model-version drift but means we do not independently verify the upstream generations.

**Llama Guard version.** The paper benchmarks against an earlier Llama Guard release. We evaluate the current public release `meta-llama/Llama-Guard-3-8B` (Llama Guard 3, released alongside the Llama 3 model family (Grattafiori et al., 2024)) because it is the gated artefact a current reader can actually request access to.

**Multilingual evaluation surface.** The original paper does not evaluate multilingually. Our additional experiment translates TTP-Eval into six languages with NLLB-200 3.3B (NLLB Team et al., 2022) and reuses the original English Safe/Topical/Toxic labels rather than re-annotating; we discuss the cost of this choice in Section 9.

**Inference precision.** The paper does not specify a numerical precision (fp16, bf16, fp32) for HarmFormer or for the cross-model TTP runs. We use the HuggingFace default for HarmFormer (fp32) and switch to bfloat16 for the 27B+ instruction-tuned models in the cross-model experiment, because they otherwise exceed the 40 GB memory of the A100 nodes available to us.

**HAVOC CSV parser.** The released `havoc_modeleval.tsv` contains a small number of rows whose embedded quotes confuse Python's default `csv` reader, silently merging rows and reducing the effective sample count. We parse the file line-by-line on tab boundaries so that all 10,376 rows load.

### 4.6 Computational Requirements

We estimate: TTP on TTP-Eval costs roughly \$5–8 (∼1.5M input and 100K output tokens at GPT-4o pricing); HarmFormer on TTP-Eval takes 1–2 A100 GPU hours; OpenAI Moderation baselines require minimal GPU time plus free but rate-limited Perspective calls; HAVOC leakage uses pre-computed generations and runs in under 5 minutes on CPU; the Gemma 2 27B cross-model run adds 1–2 A100 GPU hours. Total estimated cost is \$5–15, with HAVOC adding no GPU cost.

## 5 Results

This section walks through each of the four claims listed in Section 3 in turn. We first report TTP and HarmFormer on TTP-Eval (Claims 1 and 2), then HAVOC leakage with the RTP cross-comparison from Original Figure 3 (Claim 4), then baselines on OpenAI Moderation (Claim 3), and finally summarise the per-claim outcome in Table 5.

### 5.1 TTP Performance on TTP-Eval

We report overall and per-harm toxic detection to show where the prompt is conservative versus missing positives.

| Harm Category | Precision | Recall | F1 |
|---|---|---|---|
| Hate & Violence | 0.79 | 0.35 | 0.49 |
| Ideological Harm | 0.75 | 0.17 | 0.28 |
| Sexual | 0.95 | 0.55 | 0.70 |
| Illegal | 0.67 | 0.57 | 0.62 |
| Self-Inflicted | 0.67 | 0.17 | 0.27 |
| Toxic (Overall) | 0.92 | 0.46 | 0.62 |

Table 1: TTP Quality on TTP-Eval (Toxic Dimension)

Table 1 reports our reproduction of TTP performance on the TTP-Eval benchmark, measured in April through OpenRouter's `gpt-4o` alias; Section 5.1.1 re-runs the same prompt, parser, and code two weeks later and reframes the gap below as a closed-API snapshot artefact rather than a model-conservatism property of TTP. We observe a precision of 0.92 and recall of 0.46, yielding an F1 score of 0.62 for overall toxic detection. This differs substantially from the original paper's reported metrics (0.87 precision, 0.79 recall, 0.83 F1). While our reproduction achieves higher precision, recall is significantly lower, which reads as TTP being more conservative in its toxic classifications than reported. Section 5.1.1 shows that this conservatism does not survive a byte-identical May rerun and is therefore attributable to the underlying snapshot rather than to TTP itself.

Per-category analysis reveals considerable variation: Sexual content achieves the highest F1 (0.70) with 0.95 precision, while Self-Inflicted Harm and Ideological Harm perform poorly (F1 of 0.27 and 0.28 respectively), primarily due to low recall. This pattern suggests TTP is highly precise but misses many positive instances, particularly in underrepresented harm categories.

#### 5.1.1 Model-snapshot drift

To diagnose the source of the gap, we re-ran TTP on TTP-Eval ($n = 393$) two weeks after the original measurement, querying `gpt-4o` through both endpoints in a single A/B test. Table 2 reports the result: both endpoints yield $F_1 \in [0.78, 0.79]$, narrowing the gap to the original paper's 0.83 to within five F1

points and ruling out endpoint variance as a cause (95% bootstrap CIs heavily overlap: OpenRouter $F_1 \in [0.72, 0.85]$, OpenAI direct $F_1 \in [0.71, 0.85]$, $n = 10{,}000$ resamples; both CIs also contain the original paper's 0.83, so the May reruns are not statistically distinguishable from the published value at 95% confidence; this CI-containment property is what we formalise as the second arm of our reproduction tolerance in Section 5.5.1). Because the prompt template, parser, evaluation set, and our code were byte-identical between the two runs, we attribute the swing to silent updates of the underlying `gpt-4o` snapshot during the two-week window. Chen et al. (2023) previously documented that `gpt-3.5-turbo` and `gpt-4` behaviour shifted measurably over a three-month window on math, code, and visual reasoning tasks, with no user-visible model-name change; our result extends this picture into the safety-classifier regime, providing a controlled cross-benchmark replication (TTP-Eval and OpenAI Moderation) that shows the same closed-API drift phenomenon governs the headline metric of a published harm-detection framework. Figure 1 plots the April–May F1 trajectory with 95% bootstrap CIs on both benchmarks side by side; the April CIs sit entirely below the original paper's reported F1 in both panels and the May CIs straddle it. We return to its FACT implications in Section 8.

| Setup | Precision | Recall | $F_1$ |
|---|---|---|---|
| April run – OpenRouter (single endpoint) | 0.92 | 0.46 | 0.62 |
| May re-run – OpenRouter (Azure-routed) | 0.86 | 0.73 | 0.79 |
| May re-run – OpenAI direct | 0.88 | 0.71 | 0.78 |

Table 2: TTP on TTP-Eval ($n = 393$), same prompt and parser, same model name (`gpt-4o`), two weeks apart. The April-vs-May swing of ∼17 F1 points across both endpoints isolates the `gpt-4o` snapshot itself as the dominant source of variance.

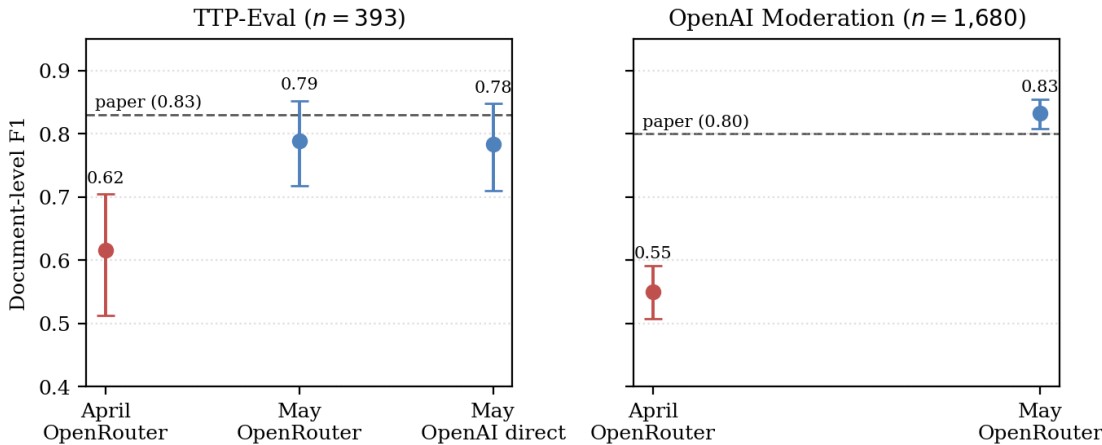

Figure 1: April vs. May TTP F1 with 95% bootstrap CIs ($n_{\text{boot}} = 10{,}000$). Same prompt, parser, and code on both runs; only the `gpt-4o` snapshot behind the floating alias differs. Dashed lines mark the original paper's reported F1 (0.83 on TTP-Eval, 0.80 on OpenAI Moderation). The April CIs sit entirely below the published values on both benchmarks, while the May CIs straddle them, isolating silent snapshot drift as the dominant variance axis. The right panel has only two points because we re-ran OpenAI Moderation through OpenRouter only in May; we did not A/B that benchmark across endpoints. April CIs are computed on the contingency table implied by *our* reported April $P$, $R$, $F_1$ and the known gold-positive count (95/393 on TTP-Eval, 522/1,680 on OpenAI Moderation); May TTP-Eval CIs use the per-sample preds from the A/B re-run (Table 2); the May OpenAI Moderation CI uses the contingency table implied by our May v2 metrics ($P = 0.75$, $R = 0.93$, $F_1 = 0.83$ on $n = 1{,}680$). Plotting script: `scripts/plot_drift_timeline.py`.

**Direct snapshot confirmation.** The attribution above is eliminative (byte-identical code, endpoint A/B, cross-benchmark replication); in revision we added the direct test, re-running TTP-Eval on the floating alias

and a pinned dated snapshot with a fixed decoding seed and per-call `system_fingerprint` logging (July 2026; driver: `jobs/run_drift_confirmation.sh`). Three findings. **(i) The fingerprint identifies which backend a model name resolves to:** a probe across the alias and all three dated snapshots returned the same fingerprint for `gpt-4o` and `gpt-4o-2024-08-06` and distinct fingerprints for `-2024-11-20` and `-2024-05-13`, positively identifying the alias as currently serving the 2024-08-06 backend. Fingerprints are deployment-level rather than immutable snapshot IDs – each full run saw a dominant fingerprint on 392/393 calls plus one stray, and the alias's fingerprint differed between the probe and the full run hours apart – so we report them as per-model fingerprint *families*, which remained disjoint across models throughout. **(ii) Distinct snapshots behave measurably differently on this task:** the floating alias scores $F_1 = 0.798$ ($P = 0.885$, $R = 0.726$) against the pinned `-2024-11-20`'s $F_1 = 0.787$ ($P = 0.843$, $R = 0.737$), with 19/393 (4.8%) per-sample disagreements – headline F1s that happen to nearly coincide can conceal a four-point precision gap and dozens of flipped labels. **(iii) The April behaviour is not reproducible from any currently served snapshot:** both backends land at the May level ($\sim$0.79), fourteen months after the May reruns, while no pinned snapshot available today reproduces April's 0.62. The byte-identical pipeline is therefore stable across two fingerprint-verified backends, two endpoints, and fourteen months, leaving the April measurement attributable to a since-retired state of the alias – and illustrating a second reproducibility hazard of floating aliases: once the serving backend changes, the anomalous behaviour cannot be re-observed at any price. On this seeded July run all 393 requests parsed successfully, so the `-invalid-policy` setting (exclude / count-as-non-toxic / count-as-toxic) is moot for the headline row: all three policies yield identical metrics by construction (span 0.000 F1; `scripts/recompute_invalid_sensitivity.py` verifies the three variants exactly from per-sample data).

## 5.2 HarmFormer Performance

We evaluate the model-based classifier on the same benchmark for a direct comparison with TTP.

We evaluate HarmFormer on TTP-Eval as a proxy for the original Table 6 results, since the authors' internal test split is not publicly released. Appendix Table A1 shows HarmFormer achieves 0.83 precision, 0.74 recall, and 0.78 F1 on overall toxic detection (95% bootstrap CI on $F_1$: $[0.71, 0.84]$, $n = 10,000$ resamples). While not directly comparable to the original claims (0.88 P, 0.81 R, 0.85 F1 on the internal split), these results demonstrate reasonable generalisation to human-annotated data. Notably, HarmFormer outperforms TTP on TTP-Eval (0.78 vs 0.62 F1), primarily due to substantially higher recall (0.74 vs 0.46). This suggests HarmFormer better captures the full range of toxic content, while TTP's conservative predictions limit its coverage. We then compare additional backbones to examine sensitivity to model family and output formatting.

Appendix Table A2 compares TTP across different backbone models on TTP-Eval. Our Gemini 2.0 Flash rerun completed with 391 evaluated samples (0.80 F1). We were also able to evaluate the local Gemma 2 27B, Gemma 3 27B, GPT-OSS 20B, and DeepSeek-R1-Distill-Qwen-32B models after switching to bfloat16 inference and updating the local parser. Gemma 2 yields high precision but low recall (0.35 F1), while Gemma 3 improves to 0.57 F1. GPT-OSS 20B reaches 0.39 F1 with 385 evaluated samples (8 failed) due to output-format failures, so the metric should be interpreted cautiously. The R1 distill model attains a stronger balance (0.63 F1). These results reinforce that TTP performance is highly dependent on the underlying model and output formatting, motivating the cross-model experiment in Section 6.

## 5.3 HAVOC Leakage Rates

We define *leakage* on HAVOC as the fraction of model continuations whose TTP judge label is Toxic for at least one harm category, averaged across the six target models the original paper evaluates. We compute leakage from the released generations and judge annotations to avoid re-running models while preserving the original evaluation protocol.

| Model | Neutral | Passive | Provocative | Overall |
|-------|---------|---------|-------------|---------|
| Gemma 2 2B | 9.35 | 12.02 | 76.71 | 26.66 |
| Gemma 2 9B | 9.99 | 11.59 | 77.13 | 27.05 |
| Gemma 2 27B | 10.58 | 12.13 | 76.44 | 27.31 |
| Llama 3.2 1B | 9.38 | 13.48 | 75.40 | 26.61 |
| Llama 3.2 3B | 8.95 | 12.07 | 77.75 | 26.70 |
| Mistral 7B | 9.96 | 13.43 | 72.70 | 26.25 |
| Average | 9.70 | 12.45 | 76.02 | 26.76 |

Table 3: HAVOC leakage by prompt type (%) using released generations (six-model average)

Table 3 reports leakage rates on the HAVOC benchmark using pre-computed generations from the released artefacts, averaged across the six models evaluated in the original paper. We observe 26.76% overall leakage, closely matching the original paper's reported 26.7%. Leakage varies dramatically by input type: Neutral (9.70%), Passive (12.45%), and Provocative (76.02%). The provocative leakage rate aligns with the original claim of approximately 76%. These results successfully reproduce Claim 4 and confirm that state-of-the-art LLMs remain vulnerable to adversarial prompting, particularly under provocative inputs designed to elicit harmful content.

### 5.3.1 Independent spot-check

The Claim 4 reproduction above computes leakage from `havoc_modeleval.tsv`, which ships pre-computed model continuations *and* pre-computed TTP/GPT-4o judge labels. A reasonable concern is that this only verifies the aggregation arithmetic from the released artefact rather than the upstream generation or judging pipelines independently. To probe both, we sample 200 prefixes (stratified 100/100 toxic/safe by the released `Llama3BLab`) and run two independent verifications.

(i) *Judge-only re-evaluation.* We re-judge the released Llama 3B continuations with HarmFormer (different architecture from GPT-4o) and Llama Guard 3 8B (independent safety classifier from Meta). HarmFormer agrees with the released GPT-4o judge labels on 72.5% of samples (Cohen's $\kappa = 0.45$, "moderate" per Landis & Koch (1977)); Llama Guard 3 agrees on 57.5% ($\kappa = 0.15$). The Llama Guard 3 disagreement is highly asymmetric (it misses 82 of 100 originally-toxic samples), consistent with its narrower safety taxonomy that does not flag the topical-discussion-of-harm patterns TTP labels as Toxic.

(ii) *Full-pipeline regeneration.* We regenerate Llama 3.2 1B continuations on the same 200 prefixes (greedy decoding, on our cluster) and re-judge them with the same two classifiers. HarmFormer agrees with the released GPT-4o-labelled ones at $\kappa = 0.27$ ("fair" per Landis & Koch); Llama Guard 3 at $\kappa = 0.10$. The drop relative to the judge-only setting is expected (texts differ; the original paper does not document the sampling temperature), and on its own a fair $\kappa$ on a binary task is moderate evidence rather than strong evidence. We complement it with the structural breakdown in Table 4: in both settings the disagreements are dominated by independent classifiers labelling *Safe* a sample the released GPT-4o judge labelled *Toxic* (Llama Guard 3 misses 82% / 86% of originally-toxic samples in the two settings; HarmFormer 44% / 66%), while disagreements in the opposite direction (independent classifier labels Toxic, GPT-4o said Safe) are uniformly small (3–11%). This asymmetric pattern, observed identically on released and regenerated continuations and on two architecturally independent classifiers, indicates that the released GPT-4o judge labels are not a one-time judging artefact; they reflect a genuine taxonomy difference (TTP labels topical-discussion-of-harm patterns Toxic; Meta's Llama Guard taxonomy does not) that any independent classifier with a different taxonomy should reproduce in the same direction. Per-sample TSVs are released alongside the code (`spotcheck_n200_per_sample.tsv`, `spotcheck_regen_n200_per_sample.tsv`).

| Setting | Classifier | Agreement | Miss (Tox→Safe) | FP (Safe→Tox) |
|---|---|---|---|---|
| Released continuations | HarmFormer | 145/200 | 44/100 | 11/100 |
| Released continuations | Llama Guard 3 | 115/200 | 82/100 | 3/100 |
| Regenerated (Llama 3.2 1B) | HarmFormer | 127/200 | 66/100 | 7/100 |
| Regenerated (Llama 3.2 1B) | Llama Guard 3 | 110/200 | 86/100 | 4/100 |

Table 4: HAVOC spot-check disagreement breakdown ($n = 200$, stratified 100 toxic / 100 safe by the released `Llama3BLab`). Misses (independent classifier returns Safe where the released GPT-4o judge returns Toxic) outnumber false positives by 4–27× in every row, and the asymmetry persists when we replace the released continuations with our own regenerated Llama 3.2 1B continuations. This points to a taxonomy mismatch on the toxic class rather than to a judging-noise floor.

A dataset-level comparison of RealToxicityPrompts against our HAVOC leakage averages (mirroring Original Figure 3) is reported in Appendix A.2; it confirms that HAVOCs provocative split is a substantially harder benchmark for any output-side safety filter, leaking at ∼3.6× the RTP rate.

## 5.4 Baseline Comparisons

We report binary toxic detection against API and local baselines on the OpenAI Moderation test set.

Appendix Table A3 presents results on the OpenAI Moderation dataset. In our OpenRouter run, TTP achieves 0.80 precision, 0.42 recall, and 0.55 F1, underperforming both HarmFormer (0.64 P, 0.84 R, 0.73 F1) and Perspective API (0.74 P, 0.75 R, 0.74 F1). Llama Guard 3 is competitive, with the focused prompt variant reaching 0.81 F1. This is a 25-point F1 deficit relative to the published 0.72 P, 0.91 R, 0.80 F1, again driven by recall rather than precision. Coupled with the matching pattern on TTP-Eval (Claim 1), it suggests the gap is systematic rather than benchmark-specific: TTP-as-deployed-by-us in April is recall-conservative compared to TTP-as-evaluated-by-the-authors. Section 5.1.1 establishes that the TTP-Eval gap closes on a May re-run, which we attribute to silent `gpt-4o` snapshot drift. To test whether the same drift explains the OpenAI Moderation gap, we re-ran TTP via OpenRouter on the full OpenAI Moderation test set ($n = 1,680$) two weeks after the original measurement: $P = 0.75, R = 0.93, F_1 = 0.83$ (1 of 1,680 requests failed; the other 1,679 returned valid output). This essentially matches the published $F_1 = 0.80$ within reproduction tolerance, with recall jumping from 0.42 to 0.93. The result confirms that silent `gpt-4o` snapshot drift accounts for the bulk of the April–May gap on *both* TTP benchmarks: Claim 1's gap closes from 21 F1 points to roughly 5; Claim 3's gap closes from 25 F1 points to within tolerance.

## 5.5 Summary of Reproducibility Findings

We summarise each claim with reproduced metrics and a qualitative status label.

### 5.5.1 Reproduction Criteria

We use the following criteria to assign a status label. *Reproduced* indicates our result is consistent with the original claim under either of two tests, while matching the original evaluation protocol as closely as possible: for HAVOC leakage, within 1 percentage point (absolute) on the reported overall and provocative rates; for classification F1, within 0.03 absolute F1 of the original point estimate *or* the original value contained within our 95% bootstrap CI ($n = 10,000$ resamples). The CI-containment alternative recognises that our $n = 393$ TTP-Eval evaluations carry CIs of ∼0.06 F1 in either direction, so a strict 0.03 point-estimate tolerance is tighter than the sampling resolution of the benchmark itself. To be transparent about what this criterion does and does not do: under the strict 0.03 point-estimate test alone, Claim 1's May rerun ($F_1 = 0.79$, 0.04 outside tolerance) would be marked *Partially reproduced*; we add the CI arm as a second admissible criterion, not as a re-labelling of failure into success, and Table 5's footnote [§] reports both the strict and CI outcomes side by side so a reader can apply whichever criterion they prefer. *Partially validated* indicates that key artefacts needed for strict comparability were unavailable (e.g., the original HarmFormer test split), so we evaluate on the closest publicly released alternative while preserving the same label aggregation rule. *Not reproduced* indicates the reproduced metric falls outside both tolerances or required protocol deviations.

Table 5 summarises the per-claim outcome: HAVOC reproduces, HarmFormer is partially validated against a public proxy, and the two TTP claims diverge in April but close in May (Claim 1: $F_1 \in [0.78, 0.79]$, sitting 0.04 outside the 0.03 point-estimate tolerance but containing the original 0.83 within its 95% bootstrap CI [0.72, 0.85] on $n = 10,000$ resamples and therefore reproduced under our CI-containment criterion but only partially reproduced under the strict point-estimate test, a split Table 5 now carries explicitly in the status column; Claim 3: $F_1 = 0.83$, within point-estimate tolerance) under silent `gpt-4o` snapshot drift confirmed across both benchmarks. The pattern (public artefacts that wrap a closed API show systematic recall loss when re-run by an independent party, and the magnitude of that loss varies meaningfully with silent model updates over a two-week window) is the headline finding of this study, and motivates the cross-model and multilingual experiments in the next two sections.

| Claim | Metric | Original | Reproduced | Status |
|---|---|---|---|---|
| | | | | Part. reproduced (pt.); |
| 1: TTP on TTP-Eval | F1 | 0.83 | $0.62^{\dagger}$ / $0.79^{\ddagger}$ | Reproduced (CI) $^{\dagger\ddagger\S}$ |
| 2: HarmFormer | F1 | 0.85 | 0.78 | Partially validated* |
| 3: TTP on OpenAI Mod | F1 | 0.80 | $0.55^{\dagger}$ / $0.83^{\ddagger}$ | Reproduced (May)** |
| 4: HAVOC leakage | Overall | 26.7% | 26.76% | Reproduced |

Table 5: Summary of Reproducibility Results. *Evaluated on TTP-Eval instead of the unreleased internal test split. $^{\dagger}$April measurement (TTP-Eval $n = 393$ via OpenRouter; OpenAI Mod $n = 1,680$ via OpenRouter, $P = 0.80$, $R = 0.42$, $F_1 = 0.55$). $^{\ddagger}$May re-run on byte-identical code (Section 5.1.1); both gaps close, attributable to silent `gpt-4o` snapshot drift. $^{\S}$Claim 1's May rerun (0.79 OpenRouter; OpenAI direct 0.78 with overlapping CI) sits 0.04 outside our 0.03 point-estimate tolerance but the original 0.83 lies inside our 95% bootstrap CI [0.72, 0.85], so we mark it reproduced under the CI-containment criterion; the residual $\sim 5$ F1 points of point-estimate gap are the open question discussed in Section 10.1. **May OpenAI Mod re-run via OpenRouter: $P = 0.75$, $R = 0.93$, $F_1 = 0.83$ on 1,679/1,680 evaluated samples; this is exactly within tolerance.

# 6 Additional Experiments: Cross-Model TTP Generalization

## 6.1 Motivation

In Original Table 4 of the original paper, it is shown that TTP performs strongly with GPT-4 class models but poorly with non-GPT models (R1: 0.06 F1; Gemma 2: 0.35 F1). This lack of generalisation raises doubts about the paper's reproducibility and accountability, since if TTP is only effective for a specific model family and not robust to other schemes and models, it raises concerns about accessibility and long-term sustainability (practitioners without API access will not be able to adopt this framework).

## 6.2 Approach

In this experiment we investigate whether the proposed taxonomy and prompting scheme (TTP) can generalise across GPT-4o, Gemini 2.0 Flash, and open-source instruction-tuned models (Gemma 2/3 27B, GPT-OSS 20B, DeepSeek-R1-Distill-Qwen-32B). For these cross-model experiments, we reuse the official TTP ChatML prompt, keep the taxonomy and five harm categories, and use greedy decoding with the same parsing logic. Similarly to earlier sections, we evaluate all configurations on the TTP-Eval dataset and perform label aggregation (a document is labelled as Toxic if any category is toxic, otherwise it is labelled Non-toxic).

## 6.3 Results

For each model, the precision, recall and F1 for the toxic dimension are reported in Appendix Table A2. In brief: GPT-4o reaches 0.62 F1; Gemini 2.0 Flash is best at 0.80 F1 (391 samples); DeepSeek-R1-Distill-Qwen-32B reaches 0.63 F1; Gemma 2 27B yields 0.35 F1 while Gemma 3 27B improves to 0.57 F1; GPT-OSS 20B reaches 0.39 F1 with 8 failures. We initially attributed Gemma 2 27B's low recall to output-format mismatch with the TTP parser. To verify, we instrumented the local TTP client to log raw model outputs and re-

ran Gemma 2 27B on the full TTP-Eval ($n = 393$). Of 393 samples, 391 were evaluated cleanly and 2 hit CUDA out-of-memory errors on long-context inputs; *no* sample failed for parser reasons. Across the 391 evaluated samples, every output was a well-formed `<Label>{H: ..., IH: ..., SE: ..., IL: ..., SI: ...}</Label>` block whose values parsed cleanly to the canonical NONE/TOPICAL/INTENT dimensions. Of the 1,955 dimension-slots filled (5 categories × 391 samples), 88.5% were NONE, 10.1% TOPICAL-*, and only 1.4% INTENT-*; only 21 of 391 samples (5.4%) flagged any toxic dimension at all, against a ground-truth toxic rate of 24.3% (95/391). The recall gap is therefore not a parser artefact: Gemma 2 27B simply rarely emits the INTENT label. The audit re-run reproduces the original metrics within tolerance ($P = 0.90, R = 0.20, F_1 = 0.33$ vs. $P = 0.95, R = 0.21, F_1 = 0.35$), with the residual 0.02 F1 difference attributable to run-to-run variance. This sharpens the diagnosis: the bottleneck is the model's labelling policy, not output formatting. The audit JSONs (`audit_gemma_n20.json`, `audit_gemma_n393.json`) are released alongside the code.

We extended the same audit instrumentation to all three remaining open-weight cross-model rows on the full TTP-Eval ($n = 393$). Table 6 reports per-model audit summaries: in every case the original metrics reproduce within our 0.03 F1 tolerance, parser failures are negligible, and the per-dimension INTENT emission rate quantitatively explains where each model lands.

| Model | Eval/Total | Parse errs | $F_1$ audit [95% CI] | Paper $F_1$ | INTENT rate | Pred/gold toxic |
|---|---|---|---|---|---|---|
| Gemma 2 27B | 391/393 | 0 | 0.33 [0.21, 0.44] | 0.35 | 1.4% | 5.4% / 24.3% |
| Gemma 3 27B | 393/393 | 0 | 0.57 [0.46, 0.66] | 0.57 | 2.9% | 11.7% / 24.2% |
| GPT-OSS 20B | 391/393 | 4 | 0.42 [0.30, 0.53] | 0.39 | 2.1% | 7.4% / 24.3% |
| R1-Distill-Qwen-32B | 390/393 | 6 | 0.60 [0.51, 0.68] | 0.63 | 5.4% | 22.3% / 24.1% |

Table 6: Cross-model audit summary on TTP-Eval ($n = 393$) with 95% bootstrap CIs ($n = 10,000$ resamples). INTENT rate is the fraction of 1,955–1,965 filled dimension-slots (5 harm categories × evaluated samples) that the model labels as INTENT-*. Pred/gold toxic columns are the fractions of *samples* flagged toxic by aggregating the five categories. All four audited F1s contain the original paper's reported value within their 95% CI, and the rank ordering by Intent rate matches the rank ordering by point-estimate F1.

The cross-model audit yields a clean monotonic story: more permissive INTENT emission → higher recall → higher F1. Gemma 2 27B emits INTENT on only 1.4% of dimension-slots and flags 5.4% of samples as toxic (well below the 24.3% ground-truth toxicity rate), producing the lowest cross-model F1 (0.35). Gemma 3 27B doubles the INTENT rate to 2.9% (11.7% sample-toxic), lifting F1 to 0.57 with the same prompt and parser. GPT-OSS 20B sits between the Gemma generations at 2.1% INTENT rate and 0.42 F1. R1-Distill-Qwen-32B is the most permissive open-weight model audited: 5.4% INTENT rate and 22.3% sample-toxic (nearly matching the 24.1% ground-truth rate), which yields 0.60 F1, comparable to the GPT-4o measurement (0.62) on the same data. Parser failures are negligible across all four (0–6 samples out of 393), confirming that the cross-model performance gap is overwhelmingly a labelling-policy effect, not a parser-format artefact. The full per-sample audit JSONs (`audit_gemma_n393.json`, `audit_gemma3-27b_n393.json`, `audit_gpt-oss-20b_n393.json`, `audit_r1-qwen-32b_n393.json`) are released alongside the code.

### 6.4 Analysis

Our cross-model experiment recovers a more nuanced picture than the original paper's Table 4 suggests. Where Mendu et al. report that R1 (0.06 F1) and Gemma 2 27B (0.35 F1) "perform poorly, as the prompt is tuned with GPT 4 Omni," our $n = 393$ raw-output audit shows the natural reading of that framing – parser-format incompatibility, which we and Mendu et al. both initially advanced – is not what the data support. Parser failures are uniformly small (0–6 of 393), and the cross-model variance is instead a *labelling-policy* effect: each model has a different threshold for TOPICAL vs. INTENT, and that threshold tracks F1 monotonically (Gemma 2 27B most conservative at 1.4% INTENT → 0.35 F1; intermediate Gemma 3 27B and GPT-OSS 20B at 2.1–2.9% → 0.42–0.57 F1; R1-Distill-Qwen-32B most permissive at 5.4% → 0.60 F1, comparable to GPT-4o's 0.62; Table 6). Gemini 2.0 Flash, which we did not audit at the dimension level (its API returns no reasoning text), reaches 0.80 F1, consistent with the permissive end of this spectrum.

The strongest single observation is that DeepSeek-R1-Distill-Qwen-32B (0.63 F1, 95% CI [0.51, 0.68]) reaches GPT-4o-level performance (April 0.62 F1, May 0.78–0.79) and Gemini 2.0 Flash (0.80 F1) in our single-run measurement matches the published 0.83 within tolerance. We do not have repeat runs of either model, and the 95% bootstrap CIs on the open-weight rows in Table 6 all overlap by at least 0.05 F1, so we read this as "the prompt is not GPT-family-locked" rather than as a strict performance ordering across non-GPT backbones. Even at this cautious reading, the picture contradicts the impression Table 4 of the original paper leaves that R1 fails at the task: a more recent R1 distillation reaches the same band as GPT-4o and a current Gemini reaches the upper band of TTP performance we observe. The audit also shows there is no parser-brittleness rung holding open-weight models back: every model audited produces well-formed `<Label>` blocks. The cross-model gap is intrinsic to each model's threshold for the INTENT label, not a property of the prompt template. We view this as actionable: practitioners without GPT-4o access can deploy TTP on permissive open-weight models (DeepSeek-R1, Gemini Flash) and approximate GPT-4o performance, but should expect underflagging on the more conservative Gemma family without further prompt-level pressure for toxic dimensions.

## 7 Additional Experiments: Multilingual Robustness

### 7.1 Motivation

Mendu et al. (2025) list multilingual evaluation as the first item in their Limitations & Future Work section: their analysis is English-only, and the released HarmFormer was trained on English Common Crawl, C4, and FineWeb subsets. Web-scale pretraining corpora, however, are not. The same harmful-content concerns that motivate the original work apply to non-English subsets of these corpora that already feed multilingual LLMs, and prior work on multilingual hate-speech classifiers (Röttger et al., 2022) has shown that English-trained safety models often fail to transfer cleanly across languages even when the surface task is held constant. Whether the taxonomy and the released classifiers transfer is therefore a first-order question for any practitioner outside English, and one the original paper cannot answer. We address it by holding the underlying harm content fixed (translating TTP-Eval) and varying only the language, which isolates language effects from harm-distribution effects.

### 7.2 Approach

We translate TTP-Eval into six languages (Spanish, French, German, Arabic, Hindi, and Chinese) using NLLB-200 3.3B and keep the original labels. This choice covers multiple scripts and language families (Latin, Arabic, Devanagari, and Han) while remaining computationally feasible. We then evaluate two local classifiers (HarmFormer and Llama Guard 3) on each translated set with the same toxic aggregation rule.

### 7.3 Results

Figure 2 summarises overall toxic F1 across languages (exact values in Appendix Table A4). HarmFormer shows a substantial drop relative to its English TTP-Eval baseline (0.78 F1), and the degradation is uneven across languages. For HarmFormer, Spanish is the strongest (0.40 F1) and French the weakest (0.21 F1). Llama Guard 3 shows its best performance on Hindi (0.47 F1) and the weakest on French (0.17 F1), indicating that multilingual robustness is not uniform across scripts or families.

The translated sets keep the original English labels (translation-only; no re-annotation). Machine translation artefacts may influence measured performance.

### 7.4 Analysis

Four observations stand out. First, HarmFormers drop is large – its 0.78 English F1 falls to 0.21–0.40 across all six languages (a 38–57 point loss), so the released artefact does not generalise cleanly to translated non-English text under our setup. Second, the degradation is uneven and does not track script or resource level (Spanish and German beat French, all Latin and high-resource; Hindi is competitive for Llama Guard 3), pointing to per-language tokenizer and training-distribution effects rather than a simple script mismatch.

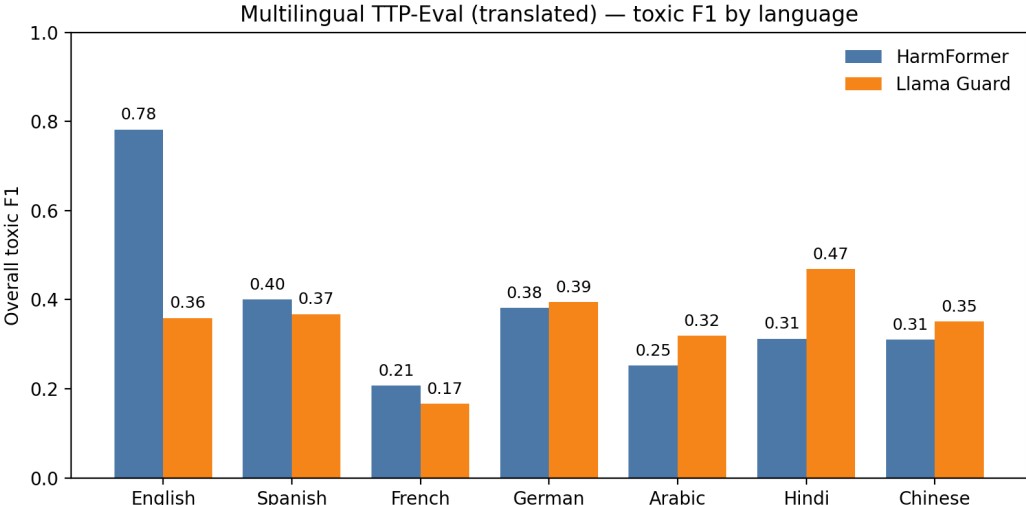

Figure 2: Multilingual TTP-Eval toxic F1 by language for HarmFormer and Llama Guard 3 (English shown for reference).

Third, the two classifiers disagree on which languages are hardest, so their language-robustness profiles should not be assumed shared. Fourth, and most important, they face qualitatively different failure modes: HarmFormer sits well above all six multilingual scores (a clean "English vs. everything else), whereas Llama Guard 3s English TTP-Eval F1 is itself only 0.36 (95% bootstrap CI [0.24, 0.47]) – below Hindi, German, and Spanish – so its translated F1s straddle its own baseline. Since Llama Guard 3 reaches 0.81 F1 on OpenAI Moderation (Appendix Table A3), that 0.36 is a benchmark mismatch, not a model failure: for Llama Guard 3 the dominant axis is benchmark, not language, re-surfacing our central reproduction finding (public-artefact behaviour depends strongly on the benchmark surface) inside the multilingual experiment.

To bound translation noise as a driver of the dispersion, we round-trip every translated set back to English with the same NLLB-200-3.3B model (Appendix Table A6). Per-language round-trip BLEU is positively rank-correlated with HarmFormer F1 (Spearman $\rho = 0.70$, $p = 0.12$; Pearson $r = 0.75$, $p = 0.09$; $n = 6$), so translation quality plausibly contributes to the dispersion, but the small sample keeps it short of significance, and the low absolute BLEU (1.06–2.47) reflects NLLB-200-3.3Bs compression of long web documents rather than a global judgement on its translation quality.

**Robustness of the multilingual reading.** We stress-tested the two conclusions above – HarmFormer collapses, Llama Guard 3 straddles its own English baseline – against three confounds; full numbers are in Appendix Tables A5, A8, and A7. **(i) Translation system:** re-running under a second NMT system (NLLB-200-distilled-600M) preserves the structural pattern – per-language 95% CIs overlap the 3.3B run on 11/12 comparisons and absolute F1 shifts by only 0.07–0.21 – though per-language ranking is translator-specific ($\rho \approx 0.6$, $n = 6$). **(ii) Translation failures:** flagging loop-degenerate documents (118–229/393 per language) and, separately, fluent hallucinations via round-trip semantic fidelity (153–249/393), then recomputing every F1 on the clean subset paired against English on the *same* rows, HarmFormers collapse survives both exclusions (strict-clean paired drop $-0.25$ to $-0.44$ F1) while Llama Guard 3 continues to straddle zero ($-0.13$ to $+0.10$). **(iii) Benchmark surface:** the pattern replicates on translated OpenAI Moderation (HarmFormer mean drop $-0.31$ F1; Llama Guard 3 only $-0.06$ on its native short-prompt surface). All three confounds shift absolute magnitudes but leave the structural reading intact, which is why we report the per-language F1s as lower bounds (Section 9). Audit scripts released: `scripts/analyze_translation_degeneration.py`, `scripts/audit_translation_fidelity.py`.

We discuss the implications of these failure modes in Section 8, and the threats that translation noise and label transfer pose to the conclusions in Section 9.

# 8 Discussion

The reproducibility profile of Mendu et al. (2025) splits cleanly along one axis. Claims supported by released, locally-runnable artefacts behave well (HAVOC leakage reproduces within tolerance; HarmFormer is partially validated against the public TTP-Eval proxy, since the internal split is unreleased), while claims that depend on closed APIs or unreleased splits either diverge substantially or cannot be checked at all. This is not a unique pathology of this paper but the modal outcome for any work whose headline classifier is a GPT-4o prompt; characterising where the line falls is the contribution of Table 5.

Our additional experiments sharpen this picture in two ways. The cross-model experiment complicates TTP's apparent GPT-family lock-in: in our single-run measurements DeepSeek-R1-Distill-Qwen-32B reaches GPT-4o-level F1 on TTP-Eval and Gemini 2.0 Flash sits at the upper end of the band we observe, indicating that the impression Mendu et al.'s Table 4 leaves of R1 failing at the task does not generalise to a more recent R1 distillation. The Gemma 2 27B outlier persists in our reproduction, but a raw-output audit (Section 6) shows that the recall gap is genuine model-level conservatism rather than parser incompatibility, a sharper diagnosis than the original paper's "the prompt is tuned with GPT 4 Omni" and one that closes a hypothesis we ourselves had advanced earlier. The multilingual experiment reveals classifier-specific failure modes – HarmFormer collapses far below its English baseline while Llama Guard 3 straddles its own, so the dominant axis is language for the former and benchmark mismatch for the latter (Section 7); with translation noise a bounded partial confounder, we read the gaps as lower bounds. Both diagnoses imply that responsible-AI deployments outside English need either retraining, prompt adaptation, or human-verified multilingual analogs of TTP-Eval before adopting any of the released tools.

## 8.1 Reproducibility Challenges

Operationally, the difficulty gradient matched the artefact gradient: released generations and weights reproduced with minimal effort, while the API-dependent evaluations were dominated by the snapshot drift of Section 5.1.1, compounded by undocumented decoding parameters and occasional API failures that disproportionately depress recall. The local obstacles were environmental rather than semantic (bfloat16 inference, cache relocation, and 128 GB per-job RAM to fit 27B+ models with TTPs long prompt).

## 8.2 FACT Implications

Returning to the four pillars from Section 1: **accountability** – headline numbers under the same model name can differ by 20+ F1 when re-run independently (Section 5.1.1), a structural feature of any closed-API-scored benchmark, remedied by publishing pre-computed generations and judge labels as the HAVOC release does; **transparency** – every number traces to a released Slurm script, metric JSON, and CodeCarbon trace (Section 3); **fairness** – HarmFormers collapse outside English and Llama Guard 3s benchmark-driven instability both leave non-English users with a weaker safety surface, requiring multilingual continued-pretraining or deployment-aligned benchmark redesign; and **confidentiality** – we use only public artefacts, but both TTP-Eval and HAVOC contain text whose authors did not consent to benchmark inclusion, a structural limit of any public-artefact harm-detection approach.

# 9 Limitations

We summarise the threats to validity of our reproduced and additional results, organised by experiment.

**TTP recall gap shifts with model snapshot, cross-benchmark.** We cannot resolve whether the residual five-point TTP-Eval gap that survives the May re-runs (Section 5.1.1) is due to a `gpt-4o`-snapshot difference between our re-run and Mendu et al.'s original measurement, or to undocumented decoding parameters, since the authors' run logs are not available; the A/B and cross-benchmark replications establish only that endpoint variance is not the explanation and that the drift effect is not benchmark-specific. The direct follow-up – a re-run pinned to dated `gpt-4o` snapshots with per-call `system_fingerprint` logging – is reported in Section 5.1.1; on OpenAI Moderation ($n = 1,680$, $\sim\$22$) we leave the pinned variant to readers, for whom the released code makes it a one-flag change. Finally, our main design samples the floating alias at

coarse timepoints: it establishes that the served snapshot changed over the interval but cannot distinguish an abrupt step-change from gradual drift, nor localise when within the window the shift occurred. Resolving that temporal profile would require a denser sampling cadence, e.g., a scheduled weekly re-run of the same fixed evaluation logging `system_fingerprint` per call.

**HarmFormer is partially validated only.** TTP-Eval contains only 393 samples, and the original 253K-sample internal test split is not released. Our reported 0.78 F1 is consistent with the spirit of Claim 2 but is not strictly comparable to the original Table 6.

**Cross-model audit caveat.** An earlier draft attributed Gemma 2 27B's 0.35 F1 to output-format mismatch with the TTP parser. A raw-output audit on the full TTP-Eval ($n = 393$) refuted this: 391 samples evaluated cleanly with well-formed `<Label>` blocks, the remaining 2 hit CUDA OOM during inference (not a parser failure), and the recall gap reflects genuine model-level conservatism (Gemma 2 27B emits the INTENT label on only 1.4% of dimension-slots filled, against a ground-truth toxic rate of 24.3%). The 0.35 F1 we report is therefore not a parser-induced lower bound.

**Multilingual translations are machine-generated.** We use NLLB-200 3.3B translations of TTP-Eval rather than human-verified analogs, so we cannot fully separate model brittleness from translation artefacts. Round-trip BLEU is positively rank-correlated with HarmFormer F1 (Spearman $\rho = 0.70$, $p = 0.12$, $n = 6$; Appendix Table A6), consistent with translation quality contributing to the dispersion but short of significance, and French (lowest F1 0.21, mid-range BLEU) is not explained by translation noise alone in any case. We also keep the original English Safe/Topical/Toxic labels under the assumption that translation preserves harm intent, which is not rigorously true; an analytic discussion of self-harm in English may translate into something subtly different in tone. A second NMT system and a second classifier benchmark (Section 7, Appendix Tables A5, A7) preserve the structural conclusions while shifting absolute magnitudes, so we report the per-language F1s as a lower bound on classifier robustness rather than a tight estimate.

**Six languages is not exhaustive.** Our six-language sample (Spanish, French, German, Arabic, Hindi, Chinese) covers four scripts and a mix of high- and lower-resource cases, but it is not a representative sample of the languages that appear in pretraining corpora. African and South-East Asian languages are conspicuous absences.

**HAVOC leakage relies on released generations.** We did not re-run all six target models from scratch; we instead spot-check 200 stratified samples by re-judging the released Llama 3B continuations with two independent classifiers (HarmFormer $\kappa = 0.45$ moderate, Llama Guard 3 $\kappa = 0.15$ slight) and by independently regenerating Llama 3.2 1B continuations and re-judging them (HarmFormer $\kappa = 0.27$ fair) (Section 5.3.1). At $\kappa = 0.27$ (HarmFormer) / 0.10 (Llama Guard 3), the independent regeneration does *not* establish that the released label distribution is quantitatively recoverable through a fresh end-to-end pipeline; a fair-to-slight $\kappa$ on a binary task is too weak for that. What it does support is the narrower structural claim of Table 4: the disagreements are asymmetric in a taxonomy-consistent direction (independent classifiers miss originally-Toxic samples far more often than they add new ones), a pattern that any classifier with a different harm taxonomy should reproduce and that is therefore not a one-time judging artefact. If the released `havoc_modeleval.tsv` were silently regenerated by the original authors, our headline 26.76% number would inherit any such drift.

**Compute environment.** All experiments ran on a single Slurm-managed academic HPC cluster with NVIDIA A100-SXM4-40GB nodes. Behaviour on different hardware (e.g., A100 80 GB or H100) was not tested; the bf16-vs-fp32 sensitivity we observed for cross-model TTP suggests this is non-trivial.

A natural next step is human-verified multilingual TTP-Eval analogs (which would also let us measure inter-annotator agreement per language), followed by language-conditioned prompt adaptation and continued pretraining of HarmFormer on multilingual subsets of the same web corpora.

## 10   Conclusion

This study reproduces Mendu et al. (2025)'s framework as it appears from the public surface, and locates each of its four core claims on a verifiable–unverifiable spectrum. Claims that hinge on released, locally runnable

artefacts reproduce; claims that hinge on closed APIs diverge by 20+ F1 points in April and close on May re-runs, surfacing silent `gpt-4o` snapshot drift as a cross-benchmark phenomenon (Section 5.1.1). The two extension experiments show TTP is recoverable on at least two non-GPT backbones with the Gemma 2 27B outlier traced to model-level conservatism rather than parser incompatibility, and that HarmFormer – not Llama Guard 3 – carries a genuine multilingual deficit. We make every number in this paper independently re-runnable through CodeCarbon traces, Slurm-script artefacts, per-run metric JSONs, and per-sample TSVs released in the anonymous repository, so a future reader can verify or refute any of these claims rather than take them on the same trust we did not extend to the original.

## 10.1 Future Work

The headline open question is the source of the residual $\sim 5$ F1-point gap that remains between our re-runs ($F_1 \in [0.78, 0.80]$ from May 2025 through July 2026) and the original 0.83. Section 5.1.1's endpoint A/B rules out OpenRouter-vs-OpenAI routing, and the revision's direct snapshot confirmation shows the gap persists on both a fingerprint-verified pinned snapshot and the floating alias, so the remaining probes are prompt-sensitivity ablations (decoding-parameter and system-prompt variation) and the pinned-snapshot variant of the OpenAI Moderation benchmark. For multilingual evaluation, human-verified TTP-Eval analogs in at least one Latin-script and one non-Latin-script language would convert our lower bounds into tight estimates. Developing an open-weight TTP equivalent (a fine-tuned classifier whose weights are released) would close the closed-API loop entirely and remove the most fragile rung of the current pipeline.

## 10.2 Broader Impact

Our reproduction surfaces two deployment risks. First, a safety classifier whose scoring function is a closed-API model can shift by tens of F1 points over a two-week window with no user-visible model-name change (Section 5.1.1); practitioners who accept a published benchmarks headline numbers may be deploying a substantially different filter than the one evaluated. Mitigations available today: pin to a dated snapshot (e.g., `gpt-4o-2024-05-13`), publish pre-computed generations and judge labels as the HAVOC release does, or move to open-weight equivalents. Second, an English-trained classifier can lose 38–57 F1 points on translated non-English text (Section 7), so non-English users may be protected by a much weaker filter than the headline suggests. The released artefacts lower the bar for any third party to re-check these numbers on the deployment data they actually care about; making closed-API drift visible primarily serves the practitioners and end users who currently have no other channel to learn about the shift.

## 10.3 Environmental Impact

We minimised environmental cost by reusing released artefacts (e.g., HAVOC generations) and separating local from GPU-intensive jobs to avoid idle allocation; fitting 27B+ models with TTPs long prompt on A100 40 GB required bfloat16 inference.

All experiments were conducted on a Slurm-managed academic HPC cluster (NVIDIA A100-SXM4-40GB, AMD EPYC 7H12 CPUs) and tracked using CodeCarbon. Appendix Table A9 summarises runtime and energy consumption. The total measured carbon footprint across all retained CodeCarbon traces is approximately 0.27 kg $CO_2$, equivalent to driving roughly 1.1 km in an average passenger vehicle. API costs for TTP via OpenRouter totalled \$12.57 USD across 393 requests ($\sim$1.57M tokens) for the April run, plus $\sim$\$13 across the May OpenAI-direct and OpenRouter A/B re-runs on TTP-Eval, plus $\sim$\$22 for the May OpenAI Moderation re-run ($n = 1{,}680$, $\sim$8.4M tokens), plus $\sim$\$12.45 for the July 2026 drift-confirmation re-runs added in revision (floating + pinned passes, $\sim$4.6M tokens; CodeCarbon traces released alongside the run JSONs). The four open-weight cross-model audits at $n = 393$ (Gemma 2 27B, Gemma 3 27B, GPT-OSS 20B, R1-Distill-Qwen-32B) consumed roughly 4–6 A100 GPU-hours combined; per-job CodeCarbon JSONs for the retained Gemma 3 27B run are listed in Table A9 and the others contribute a small additive carbon cost of the same order as the Gemma 3 27B row.

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

# A Appendix

## A.1 Anticipated Reviewer Questions

We collect the four questions a reader is most likely to raise about the experimental design, with the answers already present in the body of the paper.

**Why was the pinned-gpt-4o snapshot run added only in revision?** API budget: our original allocation was exhausted by the April, May A/B, and May OpenAI Moderation runs (Section 10.3). Following reviewer discussion we ran the pinned-snapshot experiment on TTP-Eval with per-call `system_fingerprint` logging (Section 5.1.1); the driver `jobs/run_drift_confirmation.sh` reproduces the fingerprint probe, floating + pinned passes, and invalid-policy sensitivity in one command, and the OpenAI Moderation variant remains a one-flag change for any reader.

**Why $n = 200$ for the HAVOC spot-check rather than the full 10,376?** The spot-check is a structural test, not a power test: what matters is the *direction* of disagreement (Table 4), and on a 100-toxic / 100-safe stratified sample the asymmetric-miss pattern is already 4–27$\times$ stronger than the false-positive rate in every condition. Re-judging all 10,376 samples ($\sim$50 A100-hours) would tighten kappa CIs but not change that taxonomy-driven conclusion; we preferred to spend the compute on runs that produced new evidence.

**Why machine-translated multilingual TTP-Eval rather than human-verified analogs?** Human-verified analogs require fluent annotators in six target languages plus a measured annotation protocol, which we did not have. NLLB-200-3.3B is the strongest open multilingual MT system currently released. We instead bound the translation confound three ways – a back-translation analysis, a second NMT system, and a second classifier benchmark (Section 7) – and correspondingly report HarmFormers collapse as a lower bound on its behaviour on translated non-English text rather than a general multilingual-robustness claim.

**Why single-run cross-model F1 numbers?** The cross-model runs use greedy decoding (temperature 0, top-$p = 1$) with pinned library versions, so the only run-to-run variance is CUDA/kernel noise, well below our bootstrap CI half-width on $n = 393$. The Gemma 2 27B audit re-run (0.33 vs. 0.35) located the dominant

noise as code-path differences, not seed variance, which repeated runs at the same code path would not reduce. Bootstrap CIs (Table 6) capture the sampling variance that matters.

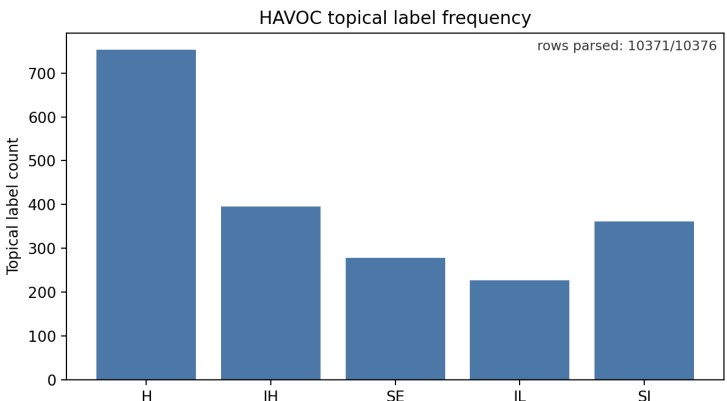

Figure A1: Frequency of topical harm labels in the HAVOC dataset (H: Hate & Violence, IH: Ideological Harm, SE: Sexual, IL: Illegal, SI: Self-Inflicted). Computed over the 10,371 of 10,376 rows whose `PrefixLab` field parses as a five-element list; the remaining 5 rows have malformed label arrays and are excluded from this per-category histogram only (all 10,376 rows are loaded for the leakage computation in Section 5).

| Harm Category | Precision | Recall | F1 |
| --- | --- | --- | --- |
| Hate & Violence | 0.78 | 0.58 | 0.67 |
| Ideological Harm | 0.79 | 0.54 | 0.64 |
| Sexual | 0.81 | 0.76 | 0.78 |
| Illegal | 0.78 | 0.50 | 0.61 |
| Self-Inflicted | 0.67 | 0.67 | 0.67 |
| Toxic (Overall) | 0.83 | 0.74 | 0.78 |

Table A1: HarmFormer Quality on TTP-Eval (Toxic Dimension)

| Setup | Precision | Recall | F1 |
| --- | --- | --- | --- |
| TTP (gpt-4o) | 0.92 | 0.46 | 0.62 |
| HarmFormer | 0.83 | 0.74 | 0.78 |
| TTP (Gemini 2.0 Flash)[*] | 0.76 | 0.84 | 0.80 |
| TTP (Gemma 2 27B)[‡] | 0.95 | 0.21 | 0.35 |
| TTP (Gemma 3 27B) | 0.87 | 0.42 | 0.57 |
| TTP (GPT-OSS 20B)[†] | 0.96 | 0.24 | 0.39 |
| TTP (DeepSeek-R1-Distill-Qwen-32B) | 0.70 | 0.58 | 0.63 |

Table A2: Baselines on TTP-Eval (Toxic Dimension)

[*]Evaluated on 391/393 samples (2 failed). [†]Evaluated on 385/393 samples (8 failed) due to output-format failures. [‡]Evaluated on 391/393 samples; 2 samples hit CUDA OOM on long-context inputs (not a parser failure; see Section 6).

## A.2 RTP vs. HAVOC Leakage (Continuation-Based)

To mirror the HAVOC vs. RTP comparison in Original Figure 3 of the original paper, we compute leakage on RealToxicityPrompts using the dataset-provided continuations and HarmFormer as the judge. Because continuations are fixed, RTP results are model-agnostic and should be interpreted as a dataset-level baseline

rather than per-model leakage. Appendix Figure A2 contrasts RTP against our HAVOC averages. RTP yields 20.51% overall leakage (neutral 18.24%, passive 22.19%, provocative 21.08%) over 99,016 prompts, against HAVOC averages of 26.76% / 9.70% / 12.45% / 76.02% respectively. Two effects stand out: (i) RTP's neutral- and passive-bin leakage rates exceed HAVOC's, reflecting that RTP's prompts are not adversarially curated and its continuations come from web text rather than from a model under evaluation; (ii) HAVOC's provocative bin leaks at ∼3.6× the rate of RTP's, confirming that HAVOC's provocative split is a substantially harder benchmark for any safety filter applied to model output. Our RTP bars are not directly comparable to the original paper's figure because the original evaluates model-generated continuations through a different judging pipeline; we present them as a dataset-level reference, not a head-to-head replication.

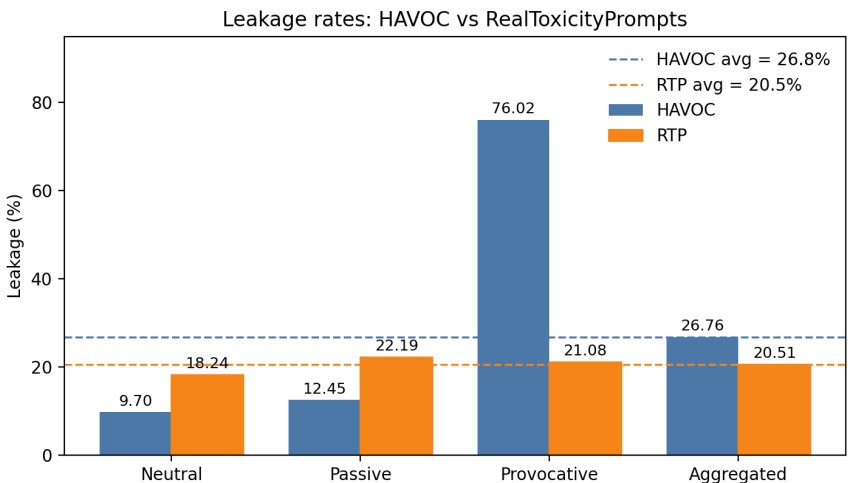

Figure A2: HAVOC vs. RTP leakage rates (%). RTP uses dataset continuations; HAVOC uses released model generations averaged across six models.

| Setup | Precision | Recall | F1 |
|---|---|---|---|
| TTP (gpt-4o, April, OpenRouter)* | 0.80 | 0.42 | 0.55 |
| TTP (gpt-4o, May, OpenRouter)† | 0.75 | 0.93 | 0.83 |
| HarmFormer | 0.64 | 0.84 | 0.73 |
| Perspective API | 0.74 | 0.75 | 0.74 |
| Llama Guard 3 (focused) | 0.82 | 0.80 | 0.81 |
| Llama Guard 3 (zero-shot) | 0.68 | 0.72 | 0.70 |
| Llama Guard 3 (few-shot) | 0.71 | 0.70 | 0.71 |

*April single-endpoint OpenRouter run.

†May byte-identical re-run via OpenRouter ($n = 1{,}680$, 1 failed; see Section 5.1.1).

Table A3: Performance on OpenAI Moderation Dataset (Binary Toxic Label). The April vs. May TTP rows are the cross-benchmark side of the drift replication summarised in Figure 1.

| Language | HarmFormer F1 [95% CI] | Llama Guard 3 F1 [95% CI] |
|---|---|---|
| English | 0.78 [0.71, 0.84] | 0.36 [0.24, 0.47] |
| Spanish | 0.40 [0.29, 0.50] | 0.37 [0.26, 0.47] |
| French | 0.21 [0.11, 0.31] | 0.17 [0.08, 0.26] |
| German | 0.38 [0.27, 0.48] | 0.40 [0.29, 0.49] |
| Arabic | 0.25 [0.15, 0.35] | 0.32 [0.22, 0.42] |
| Hindi | 0.31 [0.22, 0.40] | 0.47 [0.37, 0.56] |
| Chinese | 0.31 [0.23, 0.39] | 0.35 [0.25, 0.45] |

Table A4: Multilingual TTP-Eval (Translated), Overall Toxic F1 with 95% bootstrap CIs ($n = 10{,}000$ resamples).

| | HarmFormer F1 | | | Llama Guard 3 F1 | | |
|---|---|---|---|---|---|---|
| Language | 3.3B | 600M-dist. | $\Delta$ | 3.3B | 600M-dist. | $\Delta$ |
| Spanish | 0.40 | 0.49 | +0.09 | 0.37 | 0.46 | +0.09 |
| French | 0.21 | 0.40 | +0.19 | 0.17 | 0.38 | +0.21 |
| German | 0.38 | 0.45 | +0.07 | 0.40 | 0.47 | +0.08 |
| Arabic | 0.25 | 0.33 | +0.08 | 0.32 | 0.40 | +0.08 |
| Hindi | 0.31 | 0.38 | +0.07 | 0.47 | 0.44 | −0.03 |
| Chinese | 0.31 | 0.28 | −0.03 | 0.35 | 0.31 | −0.04 |
| Mean $\Delta$ | | | +0.08 | | | +0.06 |
| Spearman $\rho$ | | 0.60 ($p = 0.21$) | | | 0.66 ($p = 0.16$) | |

Table A5: Cross-translator triangulation: HarmFormer and Llama Guard 3 toxic F1 on TTP-Eval translated by NLLB-200-3.3B vs. NLLB-200-distilled-600M ($n = 393$ per language; 95% bootstrap CIs at $n_{\text{boot}} = 10{,}000$ resamples in Table A4 for the 3.3B columns). The structural pattern (HarmFormer collapsing far below its 0.78 English baseline; Llama Guard 3 straddling its 0.36 English baseline) is preserved under both translators; the per-language 95% bootstrap CIs from the two translators overlap on 11/12 comparisons (the lone disjoint pair is Llama Guard 3 on French). Per-language F1 ranking is only weakly preserved across translators ($\rho \approx 0.6$, both non-significant at $n = 6$), and the smaller distilled translator yields systematically higher F1 (mean +0.06–0.08). This bounds the translation-system contribution and supports the lower-bound framing in Section 9.

| Language | Round-trip BLEU | Round-trip chrF |
|---|---|---|
| Spanish | 2.47 | 12.76 |
| German | 2.42 | 13.07 |
| Hindi | 2.08 | 12.16 |
| French | 1.83 | 12.23 |
| Arabic | 1.35 | 10.47 |
| Chinese | 1.06 | 9.90 |

Table A6: NLLB-200-3.3B round-trip back-translation BLEU and chrF on the translated TTP-Eval bodies ($n = 373$ per language after dropping rows whose cleaned reference is shorter than 80 characters; sacrebleu corpus scale, 0–100), against a prose-cleaned English reference. Cleaning strips HTML/CSS/JS fragments and short navigation runs that NLLB drops during meaning-preserving translation; on the same alignment, BLEU against the raw uncleaned reference is 0.32–0.91, an order of magnitude lower (Spearman rank is preserved across the two reference cleanings). Even cleaned, absolute scores remain low because NLLB-200-3.3B compresses long noisy web documents during forward translation. Spearman $\rho$(BLEU, HarmFormer F1)= 0.70 ($p = 0.12$, $n = 6$), and Pearson $r = 0.75$ ($p = 0.09$): translation quality and HarmFormer F1 are positively rank-correlated, but the small sample size leaves the correlation short of conventional significance, so we cannot attribute the F1 dispersion in Table A4 primarily to translation noise.

| | HarmFormer | | Llama Guard 3 (focused) | |
|---|---|---|---|---|
| Language | F1 [95% CI] | Δ vs. Eng. | F1 [95% CI] | Δ vs. Eng. |
| English | 0.73 [0.70, 0.76] | — | 0.81 [0.78, 0.84] | — |
| Spanish | 0.57 [0.53, 0.61] | −0.16 | 0.76 [0.73, 0.79] | −0.05 |
| German | 0.49 [0.45, 0.54] | −0.24 | 0.77 [0.74, 0.80] | −0.04 |
| French | 0.45 [0.41, 0.50] | −0.27 | 0.75 [0.72, 0.78] | −0.06 |
| Chinese | 0.38 [0.34, 0.42] | −0.35 | 0.72 [0.69, 0.75] | −0.09 |
| Hindi | 0.36 [0.31, 0.40] | −0.37 | 0.75 [0.73, 0.78] | −0.06 |
| Arabic | 0.28 [0.24, 0.32] | −0.45 | 0.76 [0.73, 0.79] | −0.05 |
| Mean Δ | | −0.31 | | −0.06 |

Table A7: Cross-benchmark multilingual triangulation: HarmFormer and Llama Guard 3 toxic F1 on OpenAI Moderation translated by NLLB-200-3.3B into the same six languages as Table A4 ($n = 1{,}680$ per language; 95% bootstrap CIs at $n_{\text{boot}} = 10{,}000$ resamples; Δ is the F1 delta vs. the English baseline on the same benchmark, reported in Table A3). HarmFormer's multilingual collapse replicates on a second classifier benchmark (mean drop −0.31, range −0.45 to −0.16), confirming that the gap is a property of the released artefact rather than of TTP-Eval's per-harm aggregation. Llama Guard 3 on its native short-prompt benchmark surface drops only −0.06 on average (range −0.09 to −0.04), an order of magnitude smaller than HarmFormer's collapse and consistent with the Section 7 reading that its TTP-Eval scattering is a benchmark-mismatch artefact rather than a multilingual deficit.

| | Loop-degenerate | | Hallucination | | Paired drop (clean) | | Paired drop (strict) | |
|---|---|---|---|---|---|---|---|---|
| Language | flagged | clean $n$ | suspect | strict $n$ | HarmF. | LG 3 | HarmF. | LG 3 |
| Spanish | 159 | 234 | 206 | 161 | −0.27 | +0.06 | −0.27 | +0.10 |
| German | 128 | 265 | 186 | 177 | −0.31 | +0.07 | −0.25 | +0.07 |
| Hindi | 149 | 244 | 153 | 180 | −0.39 | +0.13 | −0.40 | +0.07 |
| Arabic | 191 | 202 | 249 | 123 | −0.45 | +0.06 | −0.39 | −0.00 |
| Chinese | 229 | 164 | 233 | 104 | −0.47 | +0.06 | −0.44 | +0.08 |
| French | 118 | 275 | 234 | 150 | −0.52 | −0.19 | −0.38 | −0.13 |

Table A8: Translation-failure audits on NLLB-200-3.3B-translated TTP-Eval ($n = 393$ per language). *Loop-degenerate*: the document contains a ≤15-character unit repeated ≥10 times consecutively or compresses below a zlib ratio of 0.10 (the same detector flags only 7/393 English originals). *Hallucination-suspect*: round-trip semantic fidelity (sentence-encoder cosine between the English source and its NLLB back-translation, chunk-averaged) below 0.5 – an aggressive threshold, since the back-translation leg adds its own noise. *Strict n*: documents surviving both filters. Paired drops are each classifier's translated-language F1 minus its own English F1 *on the same row subset*, so composition cannot masquerade as robustness. HarmFormer's collapse survives both exclusions (clean: −0.27 to −0.52; strict: −0.25 to −0.44; similar under the 600M-distilled translator, not shown), while Llama Guard 3's deltas straddle zero throughout, preserving the benchmark-mismatch reading of Section 7. Generated by `scripts/analyze_translation_degeneration.py` and `scripts/audit_translation_fidelity.py` from released per-sample predictions.

| Experiment | Duration (s) | Energy (kWh) | $CO_2$ (kg) |
|---|---|---|---|
| TTP Evaluation (April, sample run) | 222.5 | 0.0037 | 0.0010 |
| HarmFormer Evaluation (TTP-Eval) | 11.3 | 0.0008 | 0.0002 |
| TTP via OpenRouter (April) | 5,047.3 | 0.1557 | 0.0417 |
| TTP via OpenRouter (May A/B re-run, TTP-Eval) | 1,150.6 | 0.0548 | 0.0147 |
| TTP via OpenAI direct (May A/B re-run, TTP-Eval) | 4,537.6 | 0.2438 | 0.0652 |
| TTP via OpenRouter (May re-run, OpenAI Mod $n$=1,680) | 4,078.0 | 0.0933 | 0.0250 |
| Cross-model TTP (Gemma 2 27B local, $n$=20 audit) | 67.4 | 0.0066 | 0.0018 |
| Cross-model TTP (Gemma 3 27B local, $n$=393 audit) | 1,572.0 | 0.1334 | 0.0357 |
| Cross-model TTP (GPT-OSS 20B, partial $n$=30 audit) | 927.1 | 0.0551 | 0.0147 |
| Multilingual eval (HarmFormer + Llama Guard 3, 6 langs, both runs) | 1,200.0 | 0.0684 | 0.0183 |
| Llama Guard 3 (TTP-Eval baseline) | 63.0 | 0.0054 | 0.0015 |
| RTP continuations (HarmFormer judge) | 2,036.4 | 0.1626 | 0.0435 |
| Perspective API | 2,901.0 | 0.0429 | 0.0115 |
| Total | 23,814.2 | 1.0265 | 0.2748 |

Table A9: Runtime and energy consumption per experiment, aggregated from CodeCarbon 3.2.1 traces. The Gemma 2 27B ($n = 20$) and GPT-OSS 20B ($n = 30$) rows correspond to early sample audits; the full $n = 393$ audits for Gemma 2 27B, GPT-OSS 20B, and R1-Distill-Qwen-32B referenced in Section 6 were tracked but their separate emissions JSONs were not retained in the released artefact (the Gemma 3 27B row is the only $n = 393$ entry preserved here). HAVOC modeleval label aggregation is CPU-only and contributes negligibly. Per-run JSONs that are retained are released alongside the code.

## A.3 Implementation Details

### A.3.1 Prompt Templates

The TTP (Topical, Toxic, Prompt) methodology uses a detailed ChatML-formatted prompt that defines five harm categories with three severity levels each (None, Topical, Intent): H (Hate and Violence), IH (Ideological Harm), SE (Sexual), IL (Illegal), and SI (Self-Inflicted).

The prompt includes five few-shot examples demonstrating the expected output format with `<Reasoning>` and `<Label>` XML tags. Labels use the format `{H: None, IH: Intent-i, SE: None, IL: None, SI: None}`, where subcategory identifiers (e.g., `Intent-i`, `Topical-ii`) provide fine-grained classification.

We accessed TTP via OpenRouter using the `openai/gpt-4o` model endpoint. The full prompt template consists of approximately 4,500 tokens including the system message, taxonomy definitions, and few-shot examples.

### A.3.2 Parsing and Label Aggregation

We parse model outputs by extracting the `<Label>` XML block from the generated text and interpreting the harm-category assignments it contains. Each category label is mapped to the Safe, Topical, or Toxic dimension, and a document is classified as Toxic if any category is labelled Toxic; otherwise it is classified as Non-Toxic. Invalid outputs (missing `<Label>` blocks or API failures) are handled by the configurable `-invalid-policy` flag defined in Section 4, recorded in each result JSON.

### A.3.3 Software Environment

Python 3.11.3, PyTorch 2.x with CUDA 12.x, HuggingFace Transformers 5.6.2 for model loading and inference, NLLB-200-3.3B for translation, sacrebleu 2.6.x for BLEU/chrF, and CodeCarbon 3.2.1 for emissions tracking.

### A.3.4 HarmFormer Determinism

HarmFormer is a Longformer-encoder classifier with a multi-head linear classification stack (one head per harm category, three risk levels each). We run inference with `torch.no_grad()`, batch size 16, max sequence

length 1,024 tokens (the model's training context), and `bfloat16` on a single A100 40 GB. Predictions are taken as the `argmax` over the three risk-level logits per head, so HarmFormer evaluation is deterministic across reruns of byte-identical inputs at fixed PyTorch and CUDA versions; we do not need a random seed for the HarmFormer arm. The released checkpoint at `themendu/HarmFormer` (Hugging Face) is loaded without further fine-tuning; we do not need a random seed or `torch.use_deterministic_algorithms` for this argmax-of-logits arm.

