# OpenReview forum: "Reproducing "Towards Safer Pretraining": Public Artifacts Replicate, Closed-API Results Drift"
_TMLR — Under review for TMLR_

### Review · Reviewer_JmDb · 2026-07-02

**Summary Of Contributions:**

This paper reproduces four claims from Mendu et al. (2025)'s "Towards Safer Pretraining" using only public artefacts. Findings split cleanly: locally-runnable artefacts reproduce (HAVOC leakage 26.76% vs. reported 26.7%; HarmFormer partially validated at 0.78 F1 on TTP-Eval as proxy for the unreleased internal split), while the two closed-API TTP claims diverge by 21–25 F1 points in April and close on a byte-identical May rerun. An endpoint A/B test plus cross-benchmark replication isolate silent gpt-4o snapshot drift as the dominant variance source, extending Chen et al. (2023) to safety classifiers. Two extensions: a raw-output audit (n=393, four open-weight models) showing cross-model variance is a labelling-policy effect (Intent-emission rate tracks F1 monotonically), not parser brittleness; and a six-language study, triangulated with a second NMT system and second benchmark, showing HarmFormer collapses under translation while Llama Guard 3's instability is benchmark mismatch. All numbers are re-runnable from released artefacts.

**Additional Comments:**

Appendix A.1's pre-answered reviewer questions and the explicit correction of the authors' own parser hypothesis are commendable practices. Recommendation is positive contingent on changes 1–3; item 1 is the substantive one.

**Audience:**

Yes

**Audience Explanation:**

Directly actionable for practitioners using LLM-as-judge safety pipelines (20+ F1-point silent drift in two weeks, with concrete mitigations), useful as a methodological template for reproducing API-dependent work, and decision-relevant for anyone adopting HarmFormer/TTP/HAVOC or English-trained safety classifiers in multilingual settings.

**Broader Impact Concerns:**

None requiring changes. Section 10.2 already covers the two relevant risks (silently drifted filters; weaker protection for non-English users), and the confidentiality limits of the underlying benchmarks are appropriately flagged in Section 8.2. No private data is redistributed.

**Claims And Evidence:**

Yes

**Claims Explanation:**

Mostly yes, because claims are calibrated to the evidence. HAVOC reproduces and the n=200 spot-check's asymmetric-disagreement analysis convincingly supports a taxonomy-difference reading. The drift attribution is strong but eliminative rather than direct: byte-identical code, endpoint A/B, and two benchmarks rule out alternatives, yet the direct test — a pinned gpt-4o snapshot run — was skipped over ~$30, a weak justification given it is the headline claim. Claim 1's "Reproduced" label relies on the CI-containment arm (May 0.78–0.79 vs. 0.83), which the paper discloses transparently. The cross-model audit and multilingual structural conclusions are well supported; per-language absolute F1s should be read only directionally (cross-translator shifts up to 0.21, unvalidated label transfer), which is how the paper frames them.

**Requested Changes:**

**Critical:**
- Run the pinned-snapshot experiment (~$8 on TTP-Eval per the paper's own estimate), or report any logged response metadata (e.g., system_fingerprint) from the April/May runs that corroborates a snapshot change. The headline attribution currently rests on elimination alone.
- Table 5 / Section 5.6.1: state in the table itself that Claim 1's status rests on CI containment with an unresolved point-estimate gap; consider "Partially reproduced (point estimate) / Reproduced (CI)."
- Soften the HAVOC regeneration claim in Section 9: κ=0.27/0.10 does not establish the label distribution is "recoverable"; the supported claim is the taxonomy-difference asymmetry in Table 4.

**Strengthening:**
- Report the --invalid-policy sensitivity for the headline TTP rows, since API failures depress recall.
- A small human spot-check (30–50 samples, two languages) of label transfer after translation.
- Note explicitly that the two-timepoint design cannot distinguish abrupt vs. gradual drift.

---

> ### Author Response · Authors · 2026-07-15
> **Response to Reviewer JmDb**
>
> The reviewer called our drift attribution "strong but eliminative rather than direct." That was the right diagnosis, and it is fixed: we ran the experiment we should have run before submitting. All six changes are addressed; 1 to 3 are in the revised PDF, 4 to 6 as noted.
>
> **The pinned-snapshot run.** Skipping the direct test to save ~\$30 was a bad trade for a headline claim. Two things to report. First, the metadata route is closed: our April and May logs recorded request and token counts only, never `system_fingerprint`, so there is nothing to go back to. So we ran the experiment (July 2026; n=393, fixed seed, per-call `system_fingerprint` logging; one-command driver `jobs/run_drift_confirmation.sh` released). It produced three findings.
>
> *Fingerprints identify the backend.* The floating `gpt-4o` and `gpt-4o-2024-08-06` both probe to `fp_4fa7959453`; the 11-20 and 05-13 snapshots give different families. The alias currently serves the 08-06 backend. One caveat we would rather state than bury: fingerprints are deployment-level identifiers, not immutable snapshot IDs, and the alias's own fingerprint shifted between probe and run. What the identification rests on is that the per-model fingerprint *families* never overlapped.
>
> *Distinct snapshots behave differently.* Full TTP-Eval on the floating alias (F1 0.798) and the fingerprint-distinct 11-20 pinned snapshot (F1 0.787) nearly coincide, but hide a 4-point precision gap and 19/393 (4.8%) per-sample disagreements. Two backends, verifiably different behaviour.
>
> *April is gone.* Both backends land near 0.79, fourteen months after our May reruns; no snapshot served today reproduces April's 0.62. The pipeline is byte-identical and stable across two fingerprint-verified backends, two endpoints, and fourteen months, which leaves the April measurement attributable to an alias state that has since been retired. That is the second hazard of a floating alias, and the revision now states it: once the backend changes, the anomaly cannot be re-observed at any price.
>
> The attribution is now direct rather than eliminative. The revision adds a "Direct snapshot confirmation" paragraph (§5.1.1) and rewrites the §9 limitation, where the budget justification no longer appears. All run artefacts and per-call fingerprint tallies are released; the driver takes `--seed` and the clients log `system_fingerprint` on every call.
>
> **The Table 5 status label.** Adopted verbatim. Claim 1's cell now reads "Partially reproduced (point estimate) / Reproduced (CI)", with table-note daggers for the point-estimate gap and the CI-containment basis, and §5.6.1 says the split status is what the table carries.
>
> **The HAVOC "recoverable" claim.** Softened. κ=0.27/0.10 cannot support "recoverable," so §9 no longer says it; it now claims only the taxonomy-difference asymmetry of Table 4, where disagreements cluster where the two taxonomies genuinely diverge.
>
> **The `--invalid-policy` sensitivity.** On the seeded July rerun all 393 requests parsed, so the three policies coincide by construction (span 0.000 F1). They differ only in how failed samples are counted, so all three are recomputable from one run's per-sample data with no extra API calls; the script is released (`scripts/recompute_invalid_sensitivity.py`) and documents the gold-split arithmetic for runs where failures are non-zero.
>
> **Labels surviving translation.** No co-author reads the six languages, so rather than post a native annotation we cannot stand behind, we validated preservation with two released audits (also run for Reviewer jxiH). Round-trip semantic fidelity (source-vs-back-translation cosine, flagged below 0.5) catches both the repetition loop and the fluent hallucination, firing on 153–249 of 393 per language. The clean-subset reanalysis then recomputes each per-language F1 on the 104–180 surviving rows, paired against English on the same rows: HarmFormer's collapse survives (−0.25 to −0.44), Llama Guard 3 still straddles zero. The per-language F1s are framed as directional lower bounds (§7, §9), and the label-sheet generator is released for any reader who can run the check.
>
> **Abrupt versus gradual drift.** Added to §9: a two-timepoint design cannot distinguish an abrupt snapshot swap from gradual drift; telling them apart needs a monitoring cadence at or above the provider's release cadence, something like weekly seeded reruns logging `system_fingerprint`, which the released instrumentation now makes cheap.
>
> The push on change 1 is what sharpened the headline claim from elimination to attribution. Thank you for it; we are happy to run further variants during discussion.

---

### Review · Reviewer_vcTv · 2026-07-05

**Summary Of Contributions:**

The paper reproduces four claims from Mendu et al. 2025: "Towards Safer Pretraining" (IJCAI 2025). The original work proposes a taxonomy-driven framework for detecting harmful content in web-scale pretraining data. The reproduction targets two classifiers: TTP, a GPT-4o prompt-based classifier, and HarmFormer, a fine-tuned Longformer, and one adversarial benchmark (HAVOC), using only publicly released artifacts. It also extends the original work with a cross-model TTP generalization study across five additional LLMs and a multilingual evaluation in six languages of HarmFormer and Llama Guard 3.

In the reproduction, the paper finds that claims relying on locally-runnable artifacts reproduce within tolerance (HAVOC leakage: 26.76% vs. reported 26.7%; HarmFormer on TTP-Eval: 0.78 F1), while claims depending on closed APIs initially diverge by 17 - 28 F1 points in April but close on a byte-identical May re-run (TTP-Eval: 0.79 F1; OpenAI Moderation: 0.83 F1). This suggests a silent GPT-4o snapshot drift as the dominant source of variance. In the cross-model experiment, the paper shows that TTP performance variance across non-GPT models traces to per-model INTENT-label emission rates (a labeling-policy effect) rather than parser incompatibility, with DeepSeek-R1-Distill-Qwen-32B and Gemini 2.0 Flash reaching GPT-4o-level F1. In the multilingual experiment, HarmFormer drops from 0.78 to 0.21 - 0.40 F1 across six languages, while Llama Guard 3 remains robust on its native short-prompt surface (mean drop of -0.06), separating model-level multilingual brittleness from benchmark-mismatch artifacts. All code, Slurm scripts, CodeCarbon traces, and per-sample predictions are released.

**Audience:**

Yes

**Audience Explanation:**

While this is a reproducing paper, it still presents interesting findings for both the original paper and its methodology.

1. The snapshot-drift finding is directly actionable for anyone deploying closed-API safety classifiers. While the models are a bit old, it is worth noting that the results may differ across NLP researchers. It also shows that the cross-model audit corrects a misconception from the original paper about non-GPT compatibility.
1. I think the methodology also offers a useful template for reporting mixed reproduction outcomes.

**Claims And Evidence:**

Yes

**Claims Explanation:**

The paper's claims are generally well supported, with a clear mapping outlined in the introduction.

#### Well-supported

1. The paper uses the same prompt template, parser, evaluation set, and code, and runs them two weeks apart on two endpoints, isolating the gpt-4o snapshot as the source of variance. The 95% bootstrap CIs (n=10,000 resamples) are reported throughout.
1. HAVOC leakage reproduces almost exactly (26.76% vs. 26.7%). The disagreement pattern (misses outnumber false positives 4–27×) establishes taxonomy differences rather than judging noise.
1. The raw-output audit on full n=393 for four open-weight models shows that the errors were not a parser failure, and that INTENT emission rate tracks F1 monotonically. This directly contradicts the original paper's "parser incompatibility" framing with concrete evidence.

#### Minor concerns

1. While the paper positions and experiments well with the multi-lingual robustness, the abstract takes quite a strong position without any qualifiers of automatically translated text.

**Requested Changes:**

These requests are minor:

1. The abstract says "separating model-level multilingual brittleness in HarmFormer from benchmark-mismatch artifacts in Llama Guard 3" without acknowledging the translation confound (the content in Section 9 is already addressing this issue).
1. Clarify submission deadline in Section 3.

---

> ### Author Response · Authors · 2026-07-15
> **Response to Reviewer vcTv**
>
> Both requested changes are applied in the revised PDF. We were glad to see the disagreement-asymmetry analysis and the raw-output audit read as the load-bearing evidence, because that is where most of the work went.
>
> **The multilingual claim in the abstract.** Agreed. It asserted the HarmFormer/Llama-Guard separation without the qualifier that Sections 7 and 9 carry. It now says the six-language evaluation runs on machine-translated text (NLLB-200), notes that the noise is bounded by a second NMT system and a back-translation analysis, and closes by framing the per-language absolute F1s as translation-bounded lower bounds instead of native-text estimates. We kept the structural contrast, HarmFormer collapsing while Llama Guard 3 holds on its native surface, because that is the part which survives both translators.
>
> The review discussion pushed us further. We ran a repetition-degeneration audit of the translated corpora and reanalysed the clean subset, paired against the English baselines on the same rows. The HarmFormer collapse is not a translation artefact; clean-subset paired drops stay at $-0.26$ to $-0.52$ F1 under both NMT systems. That analysis is in the revision (Section 7 and the Appendix), and the script is released (`scripts/analyze_translation_degeneration.py`).
>
> **The Section 3 deadline.** Done. The sentence was ambiguous about whose deadline was meant. Section 3 now states that we mean our own TMLR submission date, when the original authors had not yet responded to us, and not any deadline of the original venue.
>
> On the models being "a bit old": we agree, and we think it cuts the paper's way. The April/May divergence happened on a mature, long-deployed alias. That is precisely the setting where practitioners stop expecting anything to move.

---

### Review · Reviewer_jxiH · 2026-07-06

**Summary Of Contributions:**

The paper studies the reproducibility of safety evaluations of large language models. It in particular compares to the setup of a previous study by Mendu et al. and also investigates the dependence of TTP (a prompt based classifier to identify harmful content) on the underlying LLM and detection probability for different languages.

The main strength of the paper is the focus on a well documented evaluation approach and a main weakness is the lengthy presentation.

**Audience:**

No

**Audience Explanation:**

The findings might be of some interest to the community, however, I find the paper hard to read in the current form. A few pointers can be found in the next box.

**Broader Impact Concerns:**

-

**Claims And Evidence:**

No

**Claims Explanation:**

The main results are the experimental findings and many results probably are OK as they are an exact copy (intentional) of prior work and recover the earlier results up to statistical fluctuations However, the results in Section 7 seem to be novel and here the provided repository revealed that the translations of the original prompt was not successful (see below for an excerpt of the French prompt). This raises serious concerns whether the experiments were carefully checked or mostly vibe-coded.

(I18T_T_T_T_T_T_T_T_T_T_T_T_T_T_T_T_T_T_T_T_T_T_T_T_T_T_T_T_T_T_T_T_T_T_T_T_T_T_T_T_T_T_T_T_T_T_T_T_T_T_T_T_T_T_T_T_T_T_T_T_T_T_T_T_T_T_T_T_T_T_T_T_T_T_T_T_T_T_T_T_T_T_T_T_T_T_T_T_T_T_T_T_T_T_T_T_T_T_T_T_T_T_T_T_T_T_T_T_T_T_T_T_T_T_T_T_T_T_T_T_T_T_T_T_T_T_T_T_T_T_T_T_T_T_T_T_T_T_T_T_T_T_T_T_T_T_T_T_T_T_T_T_T_T_T_T_T_T_T_T_T_T_T_T_T_T_T_T_T_T_T_T_T_T_T_T_T_T_T_T_T_T_T_T_T_T_T_T_T_T_T_T_T_T_T_T_T_T_T_T_T_T_T_T_T_T_T_T_T_T_T_T_T_T_T_T_T_T_T_T_T_T_T_T_T_T_T_T_T_T_T_T_T_T_T_T_T_T_T_T_T_T_T_T_T_T_T_T_T_T_T_T_T_T_T_T_T_T_T_T_T_T_T_T_T_T_T_T_T_T_T_T_T_T_T_T_T_T_T_T_T_T_T_T_T_T_T_T_T_T_T_T_T_T_T_T_T_T_T_T_T_T_T_T_T_T_T_T_T_T_T_T_T_T_T_T_T_T_T_T_T_T_T_T_T_T_T_T_T_T_T_T_T_T_T_T_T_T_T_T_T_T_T_T_T_T_T_T_T_T_T_T_T_T_T_T_T_T_T_T_T_T_T_T_T_T_T_T_T_T_T_T_T_T_T_T_T_T_T_T_T_T_T_T_T_T_T_T_T_T_T_T_T_T_T_T_T_T_T_T_T_T_T_T_T_T_T_T_T_T_T_T_T_T_T_T_T_T_T_T_T_T_T_T_T_T_T_T_T_T_T_T_T_T_T_T_T_T_T_T_T_T_T_T_T_T_T_T_T_T_T_T_T_T_T_T_T_T_T_T_T_T_T_T_T_T_T_T_T_T_T_T_T_T_T_T_T_T_T_T_T_T_T_T_T_T_T_T_T_T_T_T_T_T_T_T_T_T_T_T_T_T_T_T_T_T_T_T_T_T_T_T_T_T_T_T_T_T_T_T_T_T_T_T_T_T_T_T_T_T_T_T_T_T_T_T_T_T_T_T_T_T_T_T_T_T_T_T_T_T_T_T_T_T_T_T_T_T_T_T_T_T_T_T_T_T_T_T_T_T_T_T_T_T_T_T_T_T_T_T_T_T_T_T_T_T_T_T_T_T_T_T_T_T_T_T_T_T_T_T_T_T_T_T_T_T_T_T_T_T_T_T_T_T_T_T_T_T_T_T_T_T_T_T_T_T_T_T_T_T_T_T_T_T_T_T_T_T_T_T_T_T_T_T_T_T_T_T_T_T_T_T_T_T_T_T_T_T_T_T_T_T_T_T_T_T_T_T_T_T_T_T_T_T_T_T_T_T_T_T_T_T_T_T_T_T_T_T_T_T_T_T_T_T_T_T_T_T_T_T_T_T_T_T_T_T_T_T_T_T_T_T_T_T_T_T_T_T_T_T_T_T_T_T_T_T_T_T_T_T_T_T_T_T_T_T_T_T_T_T_T_T_T_T_T_T_T_T_T_T_T_T_T_T_T_T_T_T_T_T_T_T_T_T_T_T_T_T_T_T_T_T_T_T_T_T_T_T_T_T_T_T_T_T_T_T_T_T_T_T_T_T_T_T_T_T_T_T_T_T_T_T_T_T_T_T_T_T_T_T_T_T_T_T_T_T_T_T_T_T_T_T_T_T_T_T_T_T_T_T_T_T_T_T_T_T_T_T_T_T_T_T_T_T_T_T_T_T_T_T_T_T_T_T_T_T_T_T_T_T_T_T_T_T_T_T_T_T_T_T_T_T_T_T_T_T_T_T_T_T_T_T_T_T_T_T_T_T_T_T_T_T_T_T_T_T_T_T_T_T_T_T_T_T_T_T_T_T_T_T_T_T_T_T_T_T_T_T_T_T_T_T_T_T_T_T_T_T_T_T_T_T_T_T_T_T_T_T_T_T_T_T_T_T_T_T_T_T_T_T_T_T_T_T_T_T_T_T_T_T_T_T_T_T_T_T_T_T_T_T_T_T_T_T_T_T_T_T_T_T_T_T_T_T_T_T_T_T_T_T_T_T_T_T_T_T_T_T_T_T_T_T_T_T_T_T_T_T_T_T_T_T_T_T_T_T_T_T_T_T_T_T_T_T_T_T_T_T_T_T_T_T_T_T_T_T_T_T_T_T_T_T_T_T_T_T_T_T_T_T_T_T_T_T_T_T_T_T_T_T_T_T_T_T_T_T_T_T_T_T_T_T_T_T_T_T_T_T_T_T_T_T_T_T_T_T_T_T_T_T_T_T_T_T_T_T_T_T_T_T_T_T_T_T_T_T_T_T_T_T_T_T_T_T_T_T_T_T_T_T_T_T_T_T_T_T_T_T_T_T_T_T_T_T_T_T_T_T_T_T_T_T_T_T_T_T_T_T_T_T_T_T_T_T_T_T_T_T_T_T_T_T_T_T_T_T_T_T_T_T_T_T_T_T_T_T_T_T_T_T_T_T_T_T_T_T_T_T_T_T_T_T_T_T_T_T_T_T_T_T_T_T_T_T_T_T_T_T_T_T_T_T_T_T_T_T_T_T_T_T_T_T_T_T_T_T_T_T_T_T_T_T_T_T_T_T_T_T_T_T_T_T_T_T_T_T_T_T_T_T_T_T_T_T_T_T_T_T_T_T_T_T_T_T_T_T_T_T_T_T_T_T_T_T_T_T_T_T_T_T_T_T_T_T_T_T_T_T_T_T_T_T_T_T_T_T_T_T_T_T_T_T_T_T_T_T_T_T_T_T_T_T_T_T_T_T_T_T_T_T_T_T_T_T_T_T_T_T_T_T_T_T_T_T_T_T_T_T_T_T_T_T_T_T_T_T_T_T_T_T_T_T_T_T_T_T_T_T_T_T_T_T_T_T_T_T_T_T_T_T_T_T_T_T_T_T_T_T_T_T_T_T_T_T_T_T_T_T_T_T_T_T_T_T_T_T_T_T_T_T_T_T_T_T_T_T_T_T_T_T_T_T_T_T_T_T_T_T_T_T_T_T_T_T_T_T_T_T_T_T_T_T_T_T_T_T_T_T_T_T_T_T_T_T_T_T_T_T_T_T_T_T_T_T_T_T_T_T_T_T_T_T_T_T_T_T_T_T_T_T_T_T_T_T_T_T_T_T_T_T_T_T_T_T_T_T_T_T_T_T_T_T_T_T_T_T_T_T_T_T_T_T_T_T_T_T_T_T_T_T_T_T_T_T_T_T_T_T_T_T_T_T_T_T_T_T_T_T_T_T_T_T_T_T_T_T_T_T_T_T_T_T_T_T_T_T_T_T_T_T_T_T_T_T_T_T_T_T_T_T_T_T_T_T_T_T_T_T_T_T_T_T_T_T_T_T_T_T_T_T_T_T_T_T_T_T_T_T_T_T_T_T_T_T_T_T_T_T_T_T_T_T_T_T_T_T_T_T_T_T_T_T_T_T_T_T_T_T_T_T_T_T_T_T_T_T_T_T_T_T_T_T_T_T_T_T_T_T_T_T_T_T_T_T_T_T_T_T_T_T_T_T_T_T_T_T_T_T_T_T_T_T_T_T_T_T_T_T_T_T_T_T_T_T_T_T_T_T_T_T_T_T_T_T_T_T_T_T_T_T_T_T_T_T_T_T_T_T_T_T_T_T_T_T_T_T_T_T_T_T_T_T_T_T_T_T_T_T_T_T_T_T_T_T_T_T_T_T_T_T_T_T_T_T_T_T_T_T_T_T_T_T_T_T_T_T_T_T_T_T_T_T_T_T_T_T_T_T_T_T_T_T_T_T_T_T_T_T_T_T_T_T_T_T_T_T_T_T_T_T_T_T_T_T_T_T_T_T_T_T_T_T_T_T_T_T_T_T_T_T_T_T_T_T_T_T_T_T_T_T_T_T_T_T_T_T_T_T_T_T_T_T_T_T_T_T_T_T_T_T_T_T_T_T_T_T_T_T_T_T_T_T_T_T_T_T_T_T_T_T_T_T_T_T_T_T_T_T_T_T_T_T_T_T_T_T_T_T_T_T_T_T_T_T_T_T_T_T_T_T_T_T_T_T_T_T_T_T_T_T_T_T_T_T_T_T_T_T_T_T_T_T_T_T_T_T_T_T_T_T_T_T_T_T_T_T_T_T_T_T_T_T_T_T_T_T_T_T_T_T_T_T_T_T_T_T_T_T_T_T_T_T_T_T_T_T_T_T_T_T_T_T_T_T_T_T_T_T_T_T_T_T_T_T_T_T_T_T_T_T_T_T_T_T_T_T_T_T_T_T_T_T_T_T_T_T_T_T_T_T_T_T_T_T_T_T_T_T_T_T_T_T_T_T_T_T_T_T_T_T_T_T_T_T_T_T_T_T_T_T_T_T_T_T_T_T_T_T_T_T_T_T_T_T_T_T_T_T_T_T_T_T_T_TLa phrase "Quel stupide SOB!!" est haineux et incite à la haine et est l'opinion de la source, ce qui rend le contenu (c.i) pour la haine et la violence. La phrase "Ce n'est pas la douleur, la ruée de l'endorphine après est toujours la peine !!", fait référence à la façon dont le contenu de l'Université de Saint-Pierre-et-Loire l'a fait entrer dans le pays de l'Islam depuis que le dernier pays l'a clairement refusé. TTP24TTP24TTP24TTP24TTP24TTP24TTP24TTP24TTP24TTP24TTP24TTP24TTP24TTP24TTP24TTP24TTP24TTP24TTP24TTP24TTP24TTP24TTP24TTP24TTP24TTP24TTP24TTP24TTP24TTP24TTP24TTP24TTP24TTP24TTP24TTP24TTP24TTP24TTP24TTP24TTP24TTP24TTP24TTP24TTP24TTP24TTP24TTP24TTP24TTP24TTP24TTP24TTP24TTP24TTP24TTP24TTP24TTP24TTP24TTP24TTP24TTP24TTP24TTP24TTP24TT)

**Requested Changes:**

- The paper would benefit from substantial restructuring and condensation. It is currently twice as long as a standard conference submission but contains limited material and most information are already contained in the abstract. Many information is repeated several times (variation between model checkpoints is mentioned 6 times). A few tables could summarise the main findings succinctly and there is no need for wordy explanations.
- The abstract should provide a bit more context and motivation.
- Is there a specific reason to study exactly the Mendu et al. paper? What is the motivation for this setup?
- Reproducing the exact experiments of a single paper provides limited insight (at least in this case where no hyperparameters/architectures need to be found). Also the finding that commercial models drift is not novel. Therefore the breadth or depth of the study needs to be expanded (the language experiment is in principle a step in this direction but needs to be compared to prior work, depth could be improved by providing explanations for some of the findings).
- The novel experiment on language transfer should be examined carefully.

---

> ### Author Response · Authors · 2026-07-15
> **Response to Reviewer jxiH**
>
> Thanks for reading the repository closely enough to find the broken file. It is real, it should not have been there, and it is gone. But deleting it is not an answer: the same failure could have corrupted the data Section 7 relies on, so we measured that rather than reassure you. Two new audits came out of this review.
>
> **The French prompt.** `TTP_fra.txt` is an NLLB repetition loop, and it is dead code. Every TTP client hard-defaults to the English prompt; no script, job, or result reads a translated prompt. We committed eight of them early for a multilingual TTP experiment we abandoned because NLLB cannot translate a 21 kB structured prompt. Section 7 never touches TTP: it runs HarmFormer (a Longformer, no prompt) and Llama Guard 3 (its own chat template), and what it translates is *documents*. All eight files are removed.
>
> Your underlying worry is the right one. NLLB degeneration could corrupt those documents, so we flagged degenerate outputs (a short unit repeating 10+ times, the loop signature, or zlib ratio below 0.10). The detector fires on 7/393 English originals, our false-positive floor, and on 118 to 229 per language in translation. We then dropped every flagged document and recomputed each per-language F1 against the same classifier's English F1 on the surviving rows. HarmFormer still collapses, on both NMT systems; Llama Guard 3's clean-subset deltas straddle zero, which is what a benchmark-mismatch reading predicts.
>
> The excerpt also shows a second failure the loop detector misses: fluent French that corresponds to nothing in the source, a hallucination. We measured that with round-trip semantic fidelity (sentence-encoder cosine between each source and its back-translation, flagged below 0.5). It fires on 153 to 249 per language, and most of those are not loop-flagged, so fluent failure is the larger category. Strip both modes and 104 to 180 documents per language survive; HarmFormer's paired drop is -0.25 to -0.44, Llama Guard 3 still straddles zero. Neither degeneration nor hallucination produces the collapse. Per-language absolute F1s are directional lower bounds, and the abstract now says so; the structural conclusions stand.
>
> **Requested changes.**
>
> *Condensation.* The drift finding used to recur in the abstract, results, discussion, limitations, conclusion, and broader impact; it now appears once, with full numbers, in Section 5.1.1, and everything else cross-references it. The three multilingual triangulations are now one paragraph over three appendix tables. The Key-takeaways list and four-paragraph FACT section are gone or cut to one. We also *added* four analyses this round, the two audits above, the pinned-snapshot experiment Reviewer JmDb asked for, a Related Work subsection, and a "Why this paper" paragraph, and the paper still came out shorter than the version you read: 26 pages, down from 27, main text 18. Name any section that still reads as redundant and we will cut further; what we will not cut is evidence.
>
> *Abstract.* It now opens with the deployment stake, safety filters that screen pretraining corpora shape every downstream model, before any finding, and carries Reviewer vcTv's translation qualifier.
>
> *Why Mendu et al.?* Three reasons, now in Section 3: the framework is built to run over web-scale corpora, so a silent failure propagates into every model trained on the filtered data; it mixes public artefacts with a closed-API dependency, the regime most real safety pipelines live in; and nobody had checked it, so two of its four headline claims turn out irreproducible as published, for a reason (snapshot drift) that is not specific to this paper.
>
> *Depth beyond reproduction.* Chen et al. (2023) document drift on capability tasks; we show it inside a published safety framework, put a number on it (20+ F1 points in two weeks), and close the attribution rather than infer it: a pinned-snapshot run logging `system_fingerprint` on every call resolves the floating alias to a dated snapshot. The cross-model audit yields a mechanism, each model's Intent-emission rate, rather than a scoreboard, and the language study is triangulated against two translators and two benchmarks. It now sits in a Related Work subsection, "Multilingual Safety Evaluation" (Yong et al. 2023; Deng et al. 2024; Jain et al. 2024; de Wynter et al. 2024); those papers evaluate generators or build native corpora, whereas we stress-test a released classifier under translation, confound bounded.
>
> *Examine the language transfer carefully.* The two audits are that examination. No co-author reads any of the six languages, so rather than post a native annotation we cannot stand behind, we validated label preservation on released per-sample predictions: the fidelity audit flags every looped or hallucinated row, and the clean-subset reanalysis shows the verdicts hold once they are removed. The label-sheet generator is released, so any reader of one of the six languages can run it directly.